# Gradient Flow Sampler-based Distributionally Robust Optimization

**Zusen Xu** [1][2]  **Jia-Jie Zhu** [1]

## Abstract

We propose a mathematically principled PDE gradient flow framework for distributionally robust optimization (DRO). Exploiting the recent advances in the intersection of Monte Carlo sampling and statistical optimal transport, we show that our theoretical framework can be implemented as practical algorithms for sampling from worst-case distributions and, consequently, DRO. While numerous previous works have relied on dual reformulation techniques, we contribute a sound and complete gradient flow view based on SDEs or PDEs that can be used to construct new algorithms for general, potentially non-convex, losses. As an application, we solve a class of Wasserstein and entropy-regularized DRO problems using the recently-discovered Wasserstein Fisher-Rao and Stein variational gradient flows. Notably, we also show some simple reductions of our framework recover exactly previously proposed popular DRO methods, and provide new insights into their theoretical limits and optimization dynamics of DRO. Numerical studies based on stochastic gradient descent on machine learning tasks provide empirical backing for our theoretical findings.

## 1. Introduction

Distributionally robust optimization (DRO) (Delage & Ye, 2010; Kuhn et al., 2025) is a framework that aims to enhance the robustness of the solution to optimization problems. After the original Wasserstein distributionally robust optimization (DRO) works (Mohajerin Esfahani & Kuhn, 2018; Zhao & Guan, 2018; Gao & Kleywegt, 2023), many subsequent works have presented variations of problems with various numerical solutions. In particular, Sinkhorn DRO (Wang et al., 2025), also referred to as entropy-regularized

[1]KTH Royal Institute of Technology, Stockholm, Sweden [2]Weierstrass Institute for Applied Analysis and Stochastics, Berlin, Germany. Correspondence to: Zusen Xu <xu@wias-berlin.de>.

*Proceedings of the 43rd International Conference on Machine Learning*, Seoul, South Korea. PMLR 306, 2026. Copyright 2026 by the author(s).

Wasserstein DRO (Dapogny et al., 2023), demonstrates significant advantages in handling tasks with class imbalance. The entropy-regularized Wasserstein DRO problem can be formulated as a penalized optimization problem:

$$\min_{\theta \in \Theta} \max_{\rho \in \mathcal{P}} \left\{ \int \ell(\theta, y) \, d\rho(y) - \frac{1}{2\tau} W_\epsilon^2(\rho, \widehat{\rho_N}) \right\} \quad (1)$$

where $\tau > 0$ is the regularization parameter, $\widehat{\rho_N}$ is an empirical dataset, and $W_\epsilon^2(\mu, \nu) := \inf_{\gamma \in \Pi(\mu, \nu)} \{\mathbb{E}_\gamma[c] + \epsilon H(\gamma)\}$ is the entropy-regularized optimal transport (OT) divergence. Here, $\Pi(\mu, \nu)$ is the set of couplings with marginals $\mu$ and $\nu$, $c : \mathbb{R}^d \times \mathbb{R}^d \to \mathbb{R}$ is a cost function, $H(\gamma) = \int \gamma \log \gamma$ is the negative entropy, and $\epsilon > 0$ is the entropy regularization parameter. If we set $\epsilon = 0$, we recover the penalized version of the Wasserstein DRO as considered by Sinha et al. (2018). Without loss of generality, we will temporarily focus on the interesting choice of $W_\epsilon^2$, while other choices such as the KL divergences can also be straightforwardly adapted to our framework.

We observe that the inner maximization problem in (1) is equivalent to applying the entropy-regularized JKO operator (detailed in (5)) from the PDE literature(Jordan et al., 1998; Peyré, 2015) , resulting the equivalent formulation to (1)

$$\min_{\theta \in \Theta} \mathbb{E}_{y \sim \pi_Y} \ell(\theta, y), \quad \pi_Y = \epsilon\text{-JKO}_{\tau V}(\widehat{\rho_N}) \quad (2)$$

where $V := -\ell$. Exploiting this connection, our key insight is solving the inner problem of DRO (1) can be achieved by sampling from the worst-case distribution $\pi_Y$ using a gradient flow. This simplification of the problem structure provides intuition for novel algorithmic design, which we use to effortlessly obtain novel algorithms such as Wasserstein-Fisher-Rao and Stein variational gradient-based DRO algorithms. We term our methodology *Gradient Flow Sampler-based Distributionally Robust Optimization* (GF-DRO). We illustrate the main idea in Figure 1.

To obtain theoretical guarantees, we connect the stochastic approximation of the gradient w.r.t. $\theta$ for the outer DRO problem $\widehat{g} = \nabla_\theta \ell(\theta, y)$, $y \sim \pi_Y$, with the analysis of gradient flow-based samplers. Subsequently, the gradient estimate is used for the outer DRO problem via SGD-like updates, e.g., $\theta^{s+1} \leftarrow \theta^s - \eta \widehat{g}$. The bound of $\widehat{g}$ deviating from the true gradient can be used in the downstream standard optimization error bound for DRO solution.

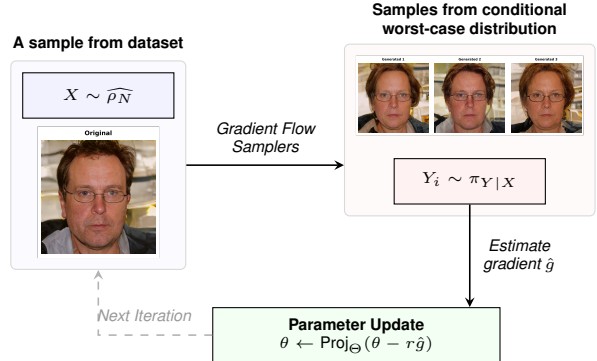

**A sample from dataset**

$X \sim \widehat{\rho_N}$

Original

*Gradient Flow Samplers*

**Samples from conditional worst-case distribution**

$Y_i \sim \pi_{Y|X}$

*Estimate gradient $\hat{g}$*

**Parameter Update**
$\theta \leftarrow \text{Proj}_\Theta(\theta - r\hat{g})$

*Next Iteration*

*Figure 1.* **GF-DRO** generates samples from the conditional worst-case distribution along the evolution of S/PDE gradient flows, which is fundamentally different from previous DRO methods. Here $\pi_{Y|X}$ denotes the conditional worst-case distribution defined by the entropic JKO operator (see Proposition 3.1). See text for detailed explanation and Section 6 for numerical experiments.

**Contributions.** Importantly, we distinguish between two different notions:

1. The DRO problem formulation with various ambiguity sets, e.g., Wasserstein DRO (Zhao & Guan, 2018; Mohajerin Esfahani & Kuhn, 2018; Gao & Kleywegt, 2023), KL-DRO (Hu & Hong, 2013; Ben-Tal et al., 2013), Sinkhorn DRO (Wang et al., 2025), Kernel DRO (Zhu et al., 2021), etc.
2. The optimization algorithm and analysis for solving the DRO problem, e.g., WRM (Sinha et al., 2018), reformulation techniques (Mohajerin Esfahani & Kuhn, 2018), SGD (Levy et al., 2020), etc.

In this paper, we do not invent new DRO problem formulations or ambiguity sets (task 1). Instead, we propose a principled mathematical framework—GF-DRO—for the analysis and design of novel DRO-solving algorithms (task 2) based on the theory of gradient flows and PDE. This framework bridges gradient-flow sampling and DRO, a connection previously missing in the literature. Unlike classical DRO approaches (Zhao & Guan, 2018; Mohajerin Esfahani & Kuhn, 2018; Gao & Kleywegt, 2023) that usually focus on specific loss functions to derive ad-hoc dual reformulations, GF-DRO provides a general framework to directly solve the variational problem (5). This allows the treatment of general, potentially non-convex losses, such as those defined by deep neural networks, for which many classical DRO methods are inapplicable. Our work can be viewed as a general-purpose *sampler-based bi-level optimization framework* (Nemirovski et al., 2009), where we invent novel DRO solvers by exploiting recently proposed gradient flow samplers, such as Wasserstein-Fisher-Rao (WFR) and Stein variational gradient (SVG) flows, to the energy functionals informed by DRO. While previous methods like WRM (Sinha et al., 2018) are fundamentally discretized ODEs, our use of S/PDEs and interacting particle systems advances

the theoretical depth of the DRO subject. For instance, Wasserstein flows relies on the log-Sobolev inequality (LSI) constant which is problematic in high dimensions, while the WFR gradient flow we introduce provides a new lane of research that may go beyond the limitations of LSI constant. Among recent sampling-based DRO efforts, FlowDRO (Xu et al., 2024) learns a deterministic transport map via Neural ODEs, and Zhu et al. (2025) analyze a JKO-like iterative scheme; in contrast, GF-DRO leverages the equivalence between the inner DRO problem and a sampling task driven by the *entropy-regularized JKO operator* (Proposition 3.1), letting us plug in any gradient flow sampler (WGF, WFR, SVG) and naturally handle entropy-regularized Wasserstein DRO. Ultimately, we demonstrate that DRO problems can be solved and analyzed in a unified, practical algorithmic framework without inventing ad-hoc reformulations.

## 2. Preliminaries

### 2.1. Gradient Systems and Their Gradient Flow Equations

The Wasserstein gradient flow (WGF) framework (Otto, 1996) was introduced into the sampling literature to provide a theoretical foundation; see Chewi et al. (2025); García Trillos & Sanz-Alonso (2018) for recent surveys. In that framework, one can write a flow equation formally as

$$\dot{\rho} = -\mathbb{G}_W(\rho)^{-1}(\rho)\frac{\delta F}{\delta \rho}[\rho] = \nabla \cdot \left(\rho \nabla \frac{\delta F}{\delta \rho}[\rho]\right) \quad (3)$$

using the inverse of the Wasserstein Riemannian metric tensor: $\mathbb{G}_W^{-1}(\rho) : T_\rho^* \mathcal{M} \to T_\rho \mathcal{M}, \xi \mapsto -\nabla \cdot (\rho \nabla \xi)$, where $\mathcal{M}$ is a space or a manifold, $T_\rho \mathcal{M}$ is the tangent space of $\mathcal{M}$ at $\rho$ and $T_\rho^* \mathcal{M}$ is the cotangent space. With those ingredients, we can formally define the gradient systems that generate gradient flow equations such as the WGF equation (3). We refer to Mielke (2023) for more details.

**Definition 2.1** (Gradient system). We refer to a tuple $(\mathcal{M}, F, \mathbb{G})$ as a gradient system. It has the gradient structure identified by:

1. a space or a manifold $\mathcal{M}$,
2. an energy functional $F$,
3. a dissipation geometry given by either: a distance metric defined on $\mathcal{M}$ or a Riemannian metric tensor $\mathbb{G}$.

### 2.2. Gradient Flow Sampler

For sampling and inference, a common choice for the energy functional is the KL divergence, i.e., $F(\rho) = \text{KL}(\rho|\pi)$. Under the Wasserstein metric, through elementary calculation, we obtain from (3) the Fokker-Planck equation (FPE)

$$\partial_t \rho = \nabla \cdot \left(\rho \nabla \log \frac{\rho}{\pi}\right) = \Delta \rho - \nabla \cdot (\rho \nabla \log \pi). \quad (4)$$

When we express the target as $\pi(x) = \frac{1}{Z}\exp(-V(x))$ where $Z$ is a normalization constant (partition function), (4) is then $\partial_t \rho = \Delta \rho + \nabla \cdot (\rho \nabla V)$. Viewed as a dynamical system, the KL divergence energy functional dissipates along (4) in the steepest descent manner. Based on Definition 2.1, we say that PDE (4) has the *gradient structure* that entails the following key ingredients:

$$\begin{cases} \text{Space}: & \text{prob. space } \mathcal{P}, \\ \text{Energy functional}: & F(\cdot) := \mathrm{KL}(\cdot|\pi), \\ \text{Dissipation Geometry}: & \text{Wasserstein metric.} \end{cases}$$

### 2.3. Entropy-regularized Wasserstein DRO

Works such as Sinha et al. (2018); Wang et al. (2025) considered DRO problems with rather general loss functions as they are based on general-purpose continuous optimization rather than DRO reformulation techniques for special losses. Wang et al. (2025) proposed a specialized algorithm solving the dual formulation of the DRO problem (1). Different from their dual approach, this paper develops a unified gradient flow sampler framework that directly samples from the primal worst-case distribution that solves the inner problem.

## 3. A Gradient Flow Framework for Sampling from Worst-case Distributions

### 3.1. Schrödinger Problem Formulation of DRO

Our starting point is the following entropy-regularized JKO (Jordan-Kinderlehrer-Otto) operator from PDE(Jordan et al., 1998; Peyré, 2015) :

$$\epsilon\text{-JKO}_{\tau V}(\rho_0) := \underset{\rho \in \mathcal{P}}{\arg\min} \left\{ \int V \, \mathrm{d}\rho + \frac{1}{2\tau} W_\epsilon^2(\rho, \rho_0) \right\} \quad (5)$$

This operation, hence the inner maximization problem of DRO, is a special case of the static Schrödinger problem with a free marginal, i.e., half bridge or one-sided bridge. We now provide a variational characterization enabling us to perform the sampling task in (2).

**Proposition 3.1.** *Solving the entropy-regularized JKO operator (5) is equivalent to solving the Schrödinger half bridge problem*

$$\min_{\substack{\Pi \in \mathcal{M}^2 \\ \int \Pi \, \mathrm{d}y = \rho_0}} \left\{ \int V \, \mathrm{d}\Pi + \frac{1}{2\tau} \int c(x,y) \, \mathrm{d}\Pi + \frac{\epsilon}{2\tau} \int \log \Pi \, \mathrm{d}\Pi \right\} \quad (6)$$

*The optimal marginal distribution of $Y$ in (6) is given by a mixture distribution, for some normalization constants $Z_x$:*

$$\pi_Y = \mathbb{E}_{x \sim \rho_0} \left[ \frac{1}{Z_x} \exp\left( -\frac{\widetilde{V}_{x,\tau}(y)}{\epsilon} \right) \right], \quad (7)$$

*where $\widetilde{V}_{x,\tau}(y) := V(y) + \frac{c(y,x)}{2\tau}$. Consequently, (6) is equivalent to the minimization of the expected KL divergence with respect to the conditional distribution*

$$\min_{\rho_{Y|X}} \mathbb{E}_{x \sim \rho_0} \mathrm{KL}\left( \rho_{Y|X=x}(y) \Big| \frac{1}{Z_x} \exp\left( -\frac{\widetilde{V}_{x,\tau}(y)}{\epsilon} \right) \right). \quad (8)$$

The optimal conditional distribution of (8), denoted by $\pi_{Y|X}$, provides the

$$\text{stochastic entropic transport map}: \quad x \mapsto \pi_{Y|X=x}(y).$$

This statement gives the overall variational structure of the DRO problem (1). We are now ready to introduce our proposed method GF-DRO in the next section.

### 3.2. Gradient Flow Sampler-based DRO

Using the variational problem (8), we define the energy functional as the KL divergence energy functional (without expectation):

$$\begin{aligned} F(\rho) := \; & \frac{\epsilon}{2\tau} \mathrm{KL}\left( \rho \, \Big| \, \frac{1}{Z_x} \exp\left( -\frac{\widetilde{V}_{x,\tau}(y)}{\epsilon} \right) \right) \\ = \; & \int V \, \mathrm{d}\rho + \frac{1}{2\tau} W_c^2(\rho, \delta_x) + \frac{\epsilon}{2\tau} \int \rho \log \rho + \text{const.} \end{aligned} \quad (9)$$

Note that the formulation on the right-hand side was also observed by Chen et al. (2022) in studying the proximal sampler.

With those ingredients, in this paper, we propose to achieve sampling from the conditional distribution $\pi_{Y|X}$ by simulating the *gradient system* $(\mathcal{P}, F, \mathbb{G})$. This results in the formal gradient flow equation:

$$\dot{\rho} = -\mathbb{G}^{-1}(\rho) \, DF(\rho). \quad (10)$$

where we can freely choose the dissipation geometry $\mathbb{G}$ for the gradient flow. Then, our central methodology is the following perspective of sampling from conditional distributions connected to the KL-minimization problem (8). Consider solving the inner maximization problem (1) via the following two-step sampling procedure:

---

**Algorithm 1** Worst-case Distribution Sampler via Gradient Flows

---

1: **Input:** an initial distribution $\rho_0$ to sample from (e.g. empirical distribution $\widehat{\rho}_N$ in data-driven DRO)
2: Sample $X \sim \rho_0$
3: Sample $Y \sim \pi_{Y|X}$ using a gradient flow (e.g. with energy functional given by (9))
4: **Output:** sample $Y$ from the worst-case distribution

---

Using the gradient flow sampler, we now propose the following general-purpose gradient flow sampler-based DRO

---

**Algorithm 2** Gradient Flow Sampler-based DRO

---

1: **Input:** Initial distribution $\rho_0$, e.g., empirical distribution $\rho_0 = \widehat{\rho_N}$, constraint set $\Theta$, $\tau, \epsilon > 0$, stepsize $r_s$.

2: **for** iteration count $s = 0, \ldots$ **do**
3:     Generate a sample from the worst-case distribution $y^s \sim \pi_Y$ by using the gradient flow sampler in Algorithm 1
4:     DRO step: $\theta^{s+1} \leftarrow \text{Proj}_\Theta(\theta^s - r_s \nabla_\theta \ell(\theta^s, y^s))$
5: **end for**

---

framework. Other straightforward variants, such as using sample average approximation (SAA) instead of stochastic approximation (SA) above, are possible.

### 3.2.1. WASSERSTEIN GRADIENT FLOW SAMPLER

As the first practical outcome of our theoretical insights, we instantiate a Wasserstein gradient flow (WGF) sampler for the entropy-regularized Wasserstein DRO problem. We consider the Wasserstein gradient system with the driving functional $F$ and the Wasserstein metric as the dissipation geometry: $(\mathcal{P}, F, W_2)$.

In sampling algorithms, this gradient flow is typically implemented by discretizing the Langevin SDE

$$dX_t = -\frac{2\tau}{\epsilon}\nabla\widetilde{V}_{x,\tau}(X_t)dt + dW_t,$$

which results in the following forward Euler discretization known as the unadjusted Langevin algorithm (ULA). Note that we use a scaled stepsize.

**Lemma 3.2.** *The forward Euler discretization of the Wasserstein gradient flow equation of the energy functional $F$ in (9) is given by the difference equation at step $t$:*

$$X_{t+1} = X_t - \eta_t \nabla\widetilde{V}_{x,\tau}(X_t) + \sqrt{\eta_t \frac{\epsilon}{\tau}}\xi_t \qquad (11)$$

*where $\xi_t$ is a standard normal random variable and $\eta_t$ is the stepsize.*

We note the stepsize scaling in front of the stochastic variable $\xi_t$ is different from the vanilla ULA update rule.

Adapting the above WGF dynamics to our GF-DRO framework in Algorithm 2 for generating samples from the worst-case distribution, we obtain the following discrete-time DRO algorithm summarized in Algorithm 3.

*Remark* 3.3. [Sinha et al. (2018)'s WRM] For the penalized Wasserstein DRO problem, i.e. $\epsilon = 0$ in the entropy-regularized Wasserstein DRO problem (1), the step in (11) specializes to the update rule:

$$X_{t+1} = X_t - \eta_t \nabla\widetilde{V}_{x,\tau}(X_t) \qquad (12)$$

---

**Algorithm 3** Entropy-regularized Wasserstein DRO via WGF

---

1: **Input:** Empirical distribution $\widehat{\rho_N}$, constraint set $\Theta$, $\tau, \epsilon > 0$, stepsize sequence $\{r_s > 0\}_{s=0}^{S-1}$, inner stepsize $\eta$, inner iterations $T$, number of samples $m$.
2: **for** $s = 0, \ldots, S-1$ **do**
3:     Sample $x^s \sim \widehat{\rho_N}$
4:     Initialize $y_0^{i,s} \leftarrow x^s, i = 1, \ldots, m$
5:     **for** $t = 0, \ldots, T-1$ **do**
6:         Sample $\xi_t^{i,s} \sim \mathcal{N}(0, I)$
7:         Update sample using (11):
        $y_{t+1}^{i,s} \leftarrow y_t^{i,s} - \eta \nabla\widetilde{V}_{x^s,\tau}(y_t^{i,s}) + \sqrt{\eta\epsilon/\tau}\xi_t^{i,s}$
8:     **end for**
9:     $\theta^{s+1} \leftarrow \text{Proj}_\Theta(\theta^s - r_s \sum_{i=1}^m \frac{1}{m}\nabla_\theta \ell(\theta^s, y_T^{i,s}))$
10: **end for**
11: **return** $\theta^S$

---

which coincides with the inner SGD step used in the WRM algorithm by Sinha et al. (2018). Hence, their WRM algorithm is a special case (with only ODE) of our GF-DRO framework in Algorithm 2.

Through the example above, we see the value of this paper's gradient flow perspective: it can easily generalize the WRM algorithm (Sinha et al., 2018) from ODE to a novel SDE-based or PDE-based algorithm for the Sinkhorn DRO problem using (11). Therefore, the gradient flow perspective is not simply a theoretical formulation – it lets us effortlessly design new algorithms to solve DRO without resorting to ad-hoc modifications of other DRO methods. Moreover, we will later go beyond the standard Wasserstein gradient flow and introduce more advanced gradient flows such as the Wasserstein Fisher-Rao (WFR) and Stein variational gradient (SVG) flows.

### 3.2.2. WASSERSTEIN FISHER-RAO FLOW SAMPLER

Consider the WFR gradient system of the energy functional $F$, i.e., the triple $(\mathcal{P}, F, \mathsf{WFR})$, where $\mathsf{WFR}$ is the Wasserstein-Fisher-Rao metric, a.k.a. the *Hellinger-Kantorovich* metric restricted to the probability space[1]. Specifically, we consider the HK/WFR gradient system associated with the reaction-diffusion PDE:

$$\partial_t \rho = \alpha \, \text{div}\left(\rho \nabla \frac{\delta F}{\delta \rho}\right) - \beta \, \rho \left(\frac{\delta F}{\delta \rho} - \int \frac{\delta F}{\delta \rho}\,\mathrm{d}\rho\right) \quad (13)$$

---

[1]Note that there are many cases of misnomer in the machine learning literature: WFR should technically be defined over the space of positive measures, not the space of probability measures; see Mielke (2025) for a historical account. The latter corresponds to the spherical Hellinger-Kantorovich metric. Although their gradient flow solutions can be easily related to each other via the mass scaling. See Mielke & Zhu (2025) for technical details.

where $\alpha, \beta > 0$ are the coefficients of the Wasserstein (transport) and Fisher-Rao (reaction) components of the HK/WFR metric, respectively. By simulating the interacting particle system associated with (13), we propose a WFR-based worst-case distribution sampler for SDRO as detailed in Algorithm 4. The primary distinction of this sampler, compared to the WGF version, is the addition of a birth-death mechanism. Crucially, this mass reallocation mechanism enables the system to concentrate its search within low-energy regions and effectively escape from local optima. See details in Appendix B.1.

### 3.2.3. Stein Variational Gradient Flow Sampler

In an attempt to simulate (4) using deterministic particle-based methods instead of SDEs, (Liu & Wang, 2016) proposed the Stein variational gradient descent (SVGD) algorithm, which is an implementable deterministic algorithmic version of (4). For a target $\pi$, at time step $t$, it updates the particle locations via the following gradient descent scheme.

$$X_{t+1}^i = X_t^i + \eta \cdot \Big(\frac{1}{m}\sum_{j=1}^m \nabla \log \pi(X_t^j) k(X_t^j, X_t^i) +$$
$$\frac{1}{m}\sum_{j=1}^m \nabla_2 k(X_t^j, X_t^i)\Big).$$

Here, $X_t^i$ represents the position of the $i$-th particle at time step $t$, $\eta$ is the stepsize, $m$ is the total number of particles. $k(\cdot, \cdot)$ is a positive definite kernel function, and $\nabla_2$ denotes the gradient with respect to the second argument of the kernel. Applying SVGD to the inner problem of DRO, we propose Algorithm 5. See details in Appendix B.2.

### 3.2.4. Rejection Sampler

We also propose Algorithm 6, which is a DRO algorithm based on rejection sampler (RGO). This approach is based on the backward step of the proximal sampler (Lee et al., 2021; Chen et al., 2022; Wibisono, 2025), as detailed in Appendix B.3. However, this method requires the loss function to be $L$-smooth and cannot be applied when $\tau$ is relatively large, which we will demonstrate in later experiments.

*Remark* 3.4 (**Dynamic interpolation: merits of a dynamic model**). Our gradient flow framework produces a principled family of distributions $\{\rho_t\}_{t\geq 0}$, where each $\rho_t$ solves a well-defined variational problem and corresponds to a specific perturbation intensity controlled by $t$. This dynamic structure has practical value: when the terminal distribution $\rho_{t_T}$ produces unrealistic samples under a "bad" hyperparameter (Figure 2), one can directly select an earlier $\rho_{t_i}$ for sample-based SGD without re-running the sampler with a different parameter (e.g. a smaller $\tau$). Unlike post-hoc interpolation between independent runs, every $\rho_t$ along the flow inherits structural guarantees from gradient flow theory, in particular

monotone energy dissipation along $F$.

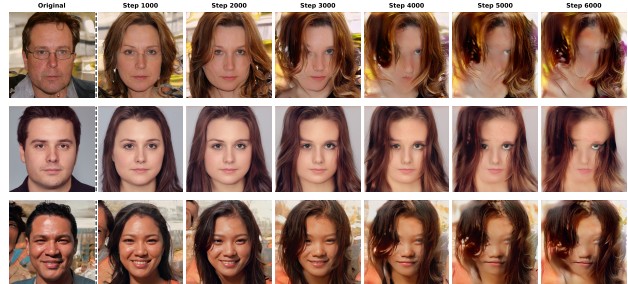

*Figure 2.* Evolution along the gradient flow under a "bad" hyperparameter. The first column displays the original images. The subsequent columns show the evolving states of the particles at intervals of 1000 steps, from Step 1000 to Step 6000.

## 4. Analysis of Gradient Flows

For algorithmic design, one paradigm is to first obtain intuition and insights from principled mathematical analysis before constructing practical numerical algorithms. While the development in machine learning research does not always follow this pattern, we argue in this paper that, by first understanding the theoretical limits of the gradient flow, we can design novel DRO algorithms that performs as expected theoretically. This is done without inventing new theoretical framework, but by leveraging the existing gradient flow and PDE analysis (Otto & Villani, 2000; Jordan et al., 1998) specialized to DRO. The discrete-time algorithmic analysis will follow in a later section.

In DRO works such as (Sinha et al., 2018) and subsequent variants, complexity analysis typically involves controlling the gradient estimation error. That is, to control the deviation from the true gradient used for DRO. For illustration, let us consider the DRO problem (1). Note that setting $\epsilon = 0$ recovers the problem of (Sinha et al., 2018). Our goal is to analyze the estimation of DRO stochastic gradient $\hat{g}$ using PDE tools. With our gradient flow sampler-based DRO algorithm, we aim to generate samples from the worst-case distribution $\pi_Y$, which is the entropic JKO solution

$$\pi_Y = \epsilon\text{-JKO}_{\tau V}(\widehat{\rho_N}) \tag{14}$$

Following the framework of Algorithm 1, for $x \sim \rho_0$, we generate samples from the worst-case distribution by sampling from the conditional distribution, $y \sim \rho_{Y|X=x}^*$. Then, for the outer DRO problem, we can use the sample-based gradient estimate $\widehat{g} := \nabla_\theta \ell(\theta, y)$ to update the parameter $\theta$. The following result uses gradient flow analysis to bound the gradient estimate error. We note that the results are stated in terms of the ideal continuous-time gradient flow, which is not yet the mixing time of the discrete-time sampler; we will detail the discrete-time algorithmic analysis in a later section. The gradient flow analysis provides a key insight and

guideline for the design of the gradient flow sampler-based DRO algorithm.

For notational simplicity, we abbreviate $\widetilde{V}_{x,\tau}$ as $\widetilde{V}$ in this section and, specifically, for the case of quadratic transport cost $c(y,x) = \|x - y\|^2$, we define the following. Recall the functional $F(\rho) = \frac{\epsilon}{2\tau} \text{KL}\left(\rho \left| \frac{1}{Z_x} \exp\left(-\frac{\widetilde{V}_{x,\tau}(y)}{\epsilon}\right)\right.\right)$ as in (9).

**Definition 4.1.** 1. **$L$-smoothness:** A differentiable function $f$ is $L$-smooth if there exists a constant $L > 0$ such that

$$\|\nabla f(x) - \nabla f(x')\| \leq L\|x - x'\|, \quad \forall x, x' \in \mathcal{X}. \tag{15}$$

2. **Semi-convexity of the regularized potential $\widetilde{V}$ :** There exists a constant $\lambda > 0$ such that

$$\nabla^2 \widetilde{V} \geq \lambda I \iff \nabla^2 V + \frac{1}{2\tau} I \geq \lambda I \tag{16}$$

3. **Gradient dominance/Polyak-Łojaciewicz (PL) inequality of $\widetilde{V}$ :** There exists a constant $\lambda > 0$ such that

$$\left|\nabla \widetilde{V}\right|^2 \geq \lambda \widetilde{V}, \quad \forall y \in \mathcal{X} \tag{17}$$

4. **Functional PL inequality over measures along GF**: There exists a constant $\lambda > 0$ such that

$$\left\|\frac{\mathrm{d}}{\mathrm{d}t} F(\rho_t)\right\|^2 \geq \lambda F(\rho_t), \tag{18}$$

which is equivalent to the $\lambda$-**logarithmic Sobolev inequality** (LSI) for $\pi = \frac{1}{Z_x} \exp\left(-\frac{\widetilde{V}_{x,\tau}(y)}{\epsilon}\right)$,

$$\int \left\|\nabla \log \frac{\rho}{\pi}\right\|^2 \mathrm{d}\rho \geq \lambda \, \text{KL}(\rho|\pi) \tag{19}$$

along the solution $\rho_t$.

It is immediate that 1. $\implies$ 2. and 1. $\implies$ 3. We note that the second condition in Definition 4.1 does not require the function $V$ itself to be convex. Hence, the DRO loss $\ell(\theta, x)$ does not necessarily need to be concave in variable $x$. In such cases, we can already obtain the geodesic convexity of $F$ in the Wasserstein space, termed displacement convexity. Letting the DRO objective $\Phi(\theta) := \max_{\rho \in \mathcal{P}} \left\{ \int \ell(\theta, z) \mathrm{d}\rho(z) - \frac{1}{2\tau} W_\epsilon^2(\rho, \widehat{\rho_N}) \right\}$, using standard analysis of gradient flows, we obtain the following ideal gradient estimate.

**Proposition 4.2.** *Denote the initial sample $x^i \sim \widehat{\rho_N}$ after running the Wasserstein gradient flow sampler for time $t$ as $y_t^i \sim \rho_{Y|X=x^i}^t$. For a fixed $\theta$, suppose that $\nabla_\theta \ell(\theta, y)$ is*

*$L_f$-Lipschitz continuous in $y$ and $\widetilde{V}_{x,\tau}(y) := -\ell(\theta, y) + \frac{1}{2\tau} c(y, x)$ is $L$-smooth and either $\lambda$-convex with $\lambda > 0$ or satisfies the $\lambda$-LSI as in Definition 4.1. Then, for sufficiently large $N$, in order to generate an $\epsilon_{grad}$-gradient estimate in expectation , i.e.,*

$$\mathbb{E}\left[\left|\frac{1}{N}\sum_{i=1}^{N} \nabla_\theta \ell(\theta, y_t^i) - \nabla_\theta \Phi(\theta)\right|\right] \leq \epsilon_{grad},$$

*the Wasserstein gradient flow sampler needs to run for time at least $t \gtrsim \mathcal{O}\left(\frac{1}{\lambda} \log \frac{L_f}{\sqrt{\lambda \epsilon_{grad}}}\right)$.*

*Remark* 4.3. The Lipschitz condition on $y \mapsto \nabla_\theta \ell(\theta, y)$ above is mild and standard, satisfied for example by losses with bounded mixed Hessian $\nabla_y \nabla_\theta \ell$. The same condition will be used in our discrete-time complexity analysis (Section 5).

Results such as Proposition 4.2 are based on the displacement convexity or PL/LSI in the Wasserstein space. The value of our general gradient flow perspective is to let us freely choose the gradient flow geometry, not confined to Wasserstein. This is not just for theoretical considerations — the estimate such as (33) depends crucially on the (PL/convexity) constant $\lambda$, which has been a major limitation of the Langevin type samplers in the literature. The potential speed-up of WFR over pure Wasserstein gradient flow comes from the Hellinger/Fisher-Rao reaction component when a suitable warm-start initialization is available; see Lu et al. (2023) and Appendix A.1 for the precise warm-start condition. It provides a simple insight for algorithmic design: the unbalanced WFR gradient flow can be used to speed up the sampling process for DRO. This insight is later validated in the numerical studies.

As noted earlier, Proposition 4.2 is aimed at providing the intuition for the behavior of the gradient flow sampler-based DRO algorithm; it is not the practical complexity estimate based on sampler's mixing time. It serves as a guideline for the design of the gradient flow sampler-based DRO algorithm using existing tools from PDE/SDE. Next, we provide optimization complexity estimate that characterizes the practical sampler-based DRO algorithm.

## 5. Optimization Complexity Analysis

The DRO problem (1) can be written as $\min_{\theta \in \Theta} \Phi(\theta)$. We analyze the computational complexity of our proposed discrete-time algorithms, aiming to find an $\epsilon_{opt}$-stationary point $\theta^S$ at the last iteration $S$, i.e., $\mathbb{E}[\|\nabla \Phi(\theta^S)\|^2] \leq \epsilon_{opt}^2$. Our analysis uses a standard framework for non-convex stochastic optimization (Ghadimi & Lan, 2013). The outer loop performs SGD on $\theta$, while the inner loop generates samples to approximate the conditional worst-case distribution. The key challenge is controlling the bias in the

stochastic gradient. We make the following assumptions regarding the objective function and the sampling process.

**Assumption 5.1.** 1. **Smoothness of the DRO objective in $\theta$ :** There exists a constant $L_\Phi > 0$ such that

$$\|\nabla\Phi(\theta_1) - \nabla\Phi(\theta_2)\| \leq L_\Phi\|\theta_1 - \theta_2\|, \forall\theta_1, \theta_2 \in \Theta$$

2. **Lipschitz continuity of the gradient oracle in $z$ :** There exists a constant $L_f > 0$ such that, for every $\theta \in \Theta$,

$$\|\nabla_\theta\ell(\theta, z_1) - \nabla_\theta\ell(\theta, z_2)\| \leq L_f\|z_1 - z_2\|, \forall z_1, z_2 \in \mathcal{Z}$$

3. **Sampler error and bounded variance:** The inner-loop sampler (Algorithm 1) generates a distribution $\widehat{\rho}_{Y|X=x}$, and $\mathbb{E}_{x\sim\widehat{\rho}_N}[W_2(\widehat{\rho}_{Y|X=x}, \pi_{Y|X=x})] \leq \delta_{\text{sample}}$. The stochastic gradient estimator $\widehat{g}$ has bounded variance $\sigma^2$, i.e., $\mathbb{E}[\|\widehat{g} - \mathbb{E}[\widehat{g}]\|^2] \leq \sigma^2$ .

Under these standard assumptions, we can bound the gradient bias and establish a general convergence result for the outer loop.

**Theorem 5.2** (Outer Loop Convergence)**.** *Let Assumptions 5.1 hold. With a stepsize $r = \sqrt{\frac{1}{SL_\Phi\sigma^2}}$, the outer loop requires $S = O(\frac{1}{\epsilon_{opt}^4})$ iterations to find an $\epsilon_{opt}$-stationary point (i.e. $\mathbb{E}[\|\nabla\Phi(\theta^S)\|^2] \leq \epsilon_{opt}^2$), provided the error $\delta_{sample}$ in Assumption 5.1.3 is controlled such that $\delta_{sample} = O(\frac{\epsilon_{opt}}{L_f})$.*

This complexity aligns with standard non-convex stochastic algorithms. Next, we derive the complexity of Algorithm 3 as an example. For the analysis of the ULA-based sampler (Algorithm 3), we introduce the following assumption on the inner target distribution.

**Assumption 5.3.** For any fixed $\theta$ and $x$, the conditional distribution $\pi_{Y|X=x}$ is $L_U$-smooth and satisfies the $\lambda_U$-log-Sobolev inequality (LSI); see Definition 4.1 for the definitions .

Assumption 5.3 provides the necessary framework to determine the number of inner ULA iterations (Vempala & Wibisono, 2019) required to achieve the sampling accuracy $\delta_{\text{sample}}$ (as specified in Assumption 5.1.3). Crucially, Assumption 5.3 relaxes the strict log-convexity assumption commonly adopted in previous works (Sinha et al., 2018; Wang et al., 2025). By combining the derived inner loop complexity analysis with the outer loop convergence rate from Theorem 5.2, we can establish the total computational complexity.

*Remark* 5.4 (Beyond smoothness)*.* Recent work shows that ULA converges under dissipativity alone (Johnston et al.), without requiring global smoothness or LSI. Since Theorem 5.2 (outer loop) is sampler-agnostic, any improved

inner-loop bound directly translates into improved total complexity. In particular, plugging in such relaxed convergence results would extend our framework to non-smooth losses (e.g., ReLU networks) more rigorously, replacing Assumption 5.3.

**Theorem 5.5** (Complexity of GF-DRO (Algorithm 3) )**.** *Under Assumptions 5.1, and 5.3, the total complexity for Algorithm 3 to find an $\epsilon_{opt}$-stationary point is $\tilde{O}\left(\frac{d}{\epsilon_{opt}^6}\right) = O\left(\frac{d}{\epsilon_{opt}^6} \cdot \log\frac{1}{\epsilon_{opt}}\right)$.*

The detailed derivations for the gradient bias, and the proofs of Theorem 5.2 and Theorem 5.5 are provided in Appendix C.

# 6. Experiments

We conduct numerical experiments on different tasks to validate our theoretical insights. Our goal is to compare the performance of four different DRO methods: Algorithm 4 (WFR-DRO), Algorithm 3 (WGF-DRO), the dual method for SDRO (Dual) (Wang et al., 2025), and WRM (Sinha et al., 2018). We also compare algorithms based on Stein Variational Gradient (SVG-DRO) (Algorithm 5) method and Restricted Gaussian Oracle (Lee et al., 2021) (RGO-DRO) (Algorithm 6) method with those methods in Appendices D.2 and D.4 . For methodological consistency across all experiments, the Dual method uses Randomized Truncation MLMC (Blanchet & Glynn, 2015) as suggested in (Wang et al., 2025), and the WRM solves its inner problem using gradient descent. Sections 6.1 and 6.2 evaluate adversarial robustness on increasingly deep convolutional architectures, while Section 6.3 demonstrates the qualitative behaviour of the WFR sampler on a high-dimensional generative latent space. All experiments were conducted on a laptop with an NVIDIA RTX A3000 GPU using Python. The code is available at `https://github.com/ZusenXu/GFS-DRO`

## 6.1. Adversarial Robustness on MNIST with LeNet-5

We evaluate our methods on MNIST using the LeNet-5 architecture in a high-dimensional, end-to-end training setting where the entire network is trained with DRO algorithms. We assess robustness using PGD attacks under both $L_2$ and $L_\infty$ norm constraints, and compare our proposed WFR-DRO and WGF-DRO ($\tau = 1.0, \epsilon = 0.05, m = 8$) against WRM and the Dual Sinkhorn DRO approach.

As shown in Tables 1 and 2, WFR-DRO consistently yields the lowest test error across all non-zero attack intensities $\Delta$, with WGF-DRO and WFR-DRO essentially tied on clean accuracy. WGF-DRO outperforms WRM under non-trivial perturbations, suggesting that entropy regularization help in non-convex landscapes; the birth–death mechanism of

*Table 1.* PGD-$L_2$ Robustness on MNIST LeNet-5 (Test Error %, mean $\pm$ std)

| Method | $\Delta = 0.00$ | $\Delta = 0.025$ | $\Delta = 0.05$ | $\Delta = 0.075$ |
|---|---|---|---|---|
| SAA | $0.92_{\pm 0.09}$ | $3.05_{\pm 0.40}$ | $10.80_{\pm 1.20}$ | $27.44_{\pm 2.86}$ |
| WRM | $0.98_{\pm 0.09}$ | $3.47_{\pm 0.39}$ | $11.66_{\pm 1.50}$ | $29.22_{\pm 3.18}$ |
| Dual | $1.00_{\pm 0.06}$ | $3.21_{\pm 0.24}$ | $9.62_{\pm 0.74}$ | $24.20_{\pm 1.82}$ |
| WGF-DRO | $0.92_{\pm 0.08}$ | $2.97_{\pm 0.36}$ | $8.56_{\pm 0.96}$ | $21.57_{\pm 1.72}$ |
| WFR-DRO | $0.92_{\pm 0.05}$ | $\mathbf{2.86}_{\pm 0.18}$ | $\mathbf{8.27}_{\pm 0.29}$ | $\mathbf{20.81}_{\pm 0.71}$ |

*Table 2.* PGD-$L_\infty$ Robustness on MNIST LeNet-5 (Test Error %, mean $\pm$ std)

| Method | $\Delta = 0.00$ | $\Delta = 0.05$ | $\Delta = 0.10$ | $\Delta = 0.15$ |
|---|---|---|---|---|
| SAA | $0.92_{\pm 0.09}$ | $3.48_{\pm 0.56}$ | $11.05_{\pm 1.98}$ | $27.70_{\pm 4.52}$ |
| WRM | $0.98_{\pm 0.09}$ | $3.58_{\pm 0.54}$ | $11.54_{\pm 2.25}$ | $29.18_{\pm 5.13}$ |
| Dual | $1.00_{\pm 0.06}$ | $3.13_{\pm 0.30}$ | $8.86_{\pm 1.07}$ | $21.29_{\pm 2.88}$ |
| WGF-DRO | $0.92_{\pm 0.08}$ | $2.81_{\pm 0.43}$ | $7.57_{\pm 1.27}$ | $17.79_{\pm 2.72}$ |
| WFR-DRO | $0.92_{\pm 0.05}$ | $\mathbf{2.68}_{\pm 0.23}$ | $\mathbf{7.41}_{\pm 0.56}$ | $\mathbf{17.30}_{\pm 1.57}$ |

WFR provides further improvement over WGF-DRO and outperforms the Dual baseline as well. WRM performs on par with SAA (within one standard deviation), which is expected in the non-convex setting(Sinha et al., 2018). For a complementary comparison on the convex feature-extracted CIFAR-10 setting, including RGO-DRO and SVG-DRO, see Appendix D.2.

## 6.2. Adversarial Robustness on CIFAR-10 with ResNet-18

We further validate the scalability of our methods in high-dimensional tasks by fine-tuning a pretrained ResNet-18 end-to-end on CIFAR-10. The DRO hyperparameters are $\tau = 0.1$ and $\epsilon = 0.05$ for Dual method, WGF-DRO and WFR-DRO. To align the comparison between WRM and methods based on SDRO, we set $\tau = 0.1$ for WRM. And $m = 8$ particles are used in WGF-DRO and WFR-DRO. The inner sampler uses step size $0.01$ for $20$ iterations across all methods. For the Dual baseline, we use the Randomized Truncation MLMC estimator (Blanchet & Glynn, 2015) with maximum truncation level $4$. All methods are trained for 20 epochs with batch size 256.

Tables 3 and 4 report the mean $\pm$ std of PGD-$L_2$ and PGD-$L_\infty$ test error under different attack intensities over 5 seeds. WFR-DRO achieves the lowest error across all non-zero attack intensities under both norms. WGF-DRO comes second, confirming that replacing the deterministic ODE with SDE alone already bring most of the robustness benefit, while WFR-DRO show an additional reduction over WGF-DRO. The Dual baseline performs well compared to WRM, suggesting that entropy regularization helps in this non-convex regime. WRM provides marginal improvement over SAA, consistent with WRM's reliance on convexity (Sinha et al., 2018) that fails to hold here. Detailed experimental settings, wall-clock timing, outer-loop convergence diagnostics, and a $\tau$-sensitivity analysis can be found in Appendix D.3.

## 6.3. Human Face Classification

In this section, we apply the WFR gradient flow sampler to the 512-dimensional latent space of a pre-trained ALAE (Pidhorskyi et al., 2020) model to investigate how

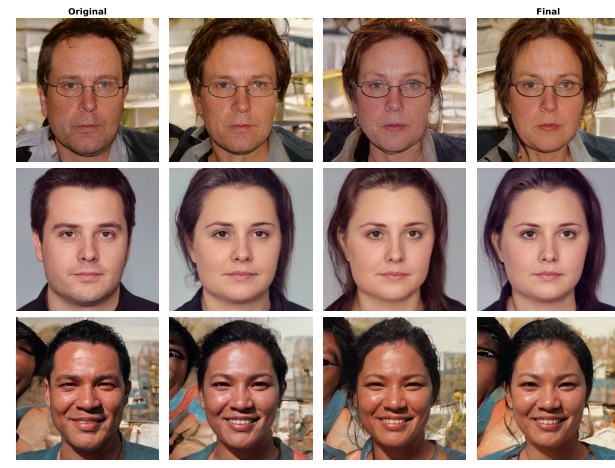

*(a)* Semantic transitions in the latent space

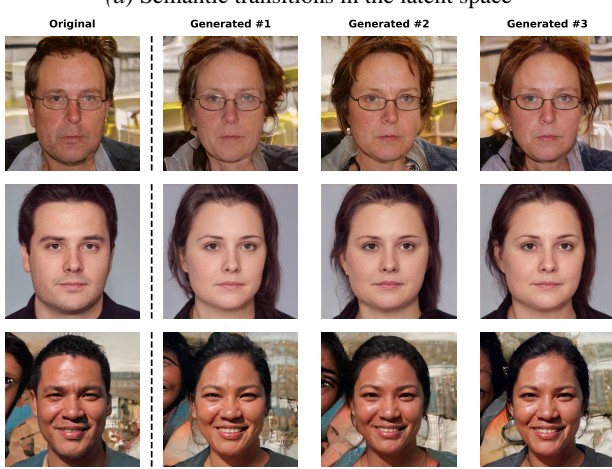

*(b)* Comparison of sample diversity

*Figure 3.* **Performance and diversity of the WFR sampler. (a)** Semantic transitions. Shown from left to right are the original images, intermediate samples at $T = 100$ and $T = 200$, and the final result at $T = 300$. **(b)** Sample diversity comparison. The first column displays the original images, while the following three columns show the three highest-weighted samples generated at $T = 300$. Unlike deterministic methods that are restricted to a single adversarial solution, the WFR sampler successfully generates multiple distinct samples from the conditional worst-case distribution.

the sampler identifies the worst-case distribution for high-dimensional image data. We employ a linear logistic regres-

*Table 3.* PGD-$L_2$ Robustness on CIFAR-10 with ResNet-18 (Test Error %, mean $\pm$ std)

| Method | $\Delta = 0.00$ | $\Delta = 0.005$ | $\Delta = 0.01$ | $\Delta = 0.015$ |
|---|---|---|---|---|
| SAA | $6.35_{\pm 0.77}$ | $25.79_{\pm 1.06}$ | $55.38_{\pm 1.02}$ | $78.32_{\pm 0.76}$ |
| WRM | $6.27_{\pm 0.55}$ | $22.27_{\pm 0.49}$ | $47.76_{\pm 1.80}$ | $71.24_{\pm 2.28}$ |
| Dual | $14.12_{\pm 0.99}$ | $19.21_{\pm 1.10}$ | $25.33_{\pm 1.44}$ | $32.00_{\pm 1.67}$ |
| WGF-DRO | $12.09_{\pm 0.75}$ | $16.47_{\pm 0.69}$ | $21.81_{\pm 0.88}$ | $27.70_{\pm 1.06}$ |
| WFR-DRO | $11.10_{\pm 0.60}$ | $\mathbf{15.39}_{\pm 0.73}$ | $\mathbf{20.63}_{\pm 0.81}$ | $\mathbf{26.73}_{\pm 0.72}$ |

*Table 4.* PGD-$L_\infty$ Robustness on CIFAR-10 ResNet-18 (Test Error %, mean $\pm$ std)

| Method | $\Delta = 0.00$ | $\Delta = 0.005$ | $\Delta = 0.01$ | $\Delta = 0.015$ |
|---|---|---|---|---|
| SAA | $6.35_{\pm 0.77}$ | $36.34_{\pm 1.31}$ | $72.46_{\pm 1.12}$ | $90.67_{\pm 0.79}$ |
| WRM | $6.27_{\pm 0.55}$ | $31.26_{\pm 0.88}$ | $65.07_{\pm 2.12}$ | $86.35_{\pm 1.57}$ |
| Dual | $14.12_{\pm 0.99}$ | $21.42_{\pm 1.20}$ | $30.19_{\pm 1.66}$ | $39.74_{\pm 1.86}$ |
| WGF-DRO | $12.09_{\pm 0.75}$ | $18.48_{\pm 0.74}$ | $26.18_{\pm 1.11}$ | $35.02_{\pm 1.13}$ |
| WFR-DRO | $11.10_{\pm 0.60}$ | $\mathbf{17.31}_{\pm 0.76}$ | $\mathbf{25.18}_{\pm 0.88}$ | $\mathbf{34.06}_{\pm 0.83}$ |

sion classifier trained on the FFHQ dataset to classify gender attributes in the latent space; the gradient of this classifier then serves as the potential function. This experiment is qualitative: we use it to visually demonstrate the diversity of worst-case samples generated by the WFR sampler, complementing the quantitative robustness evidence in Sections 6.1 and 6.2.

The WFR sampler is configured with parameters $T = 300$, $\eta = 0.05$, $\epsilon = 0.2$, and $\tau = 5$. As shown in Figure 3a, the sampler produces continuous semantic transitions while maintaining key identity details such as pose and expression. Unlike deterministic baselines such as WRM, which converge to a single adversarial solution, the WFR sampler generates diverse samples. This diversity is illustrated in Figure 3b, where the top-3 weighted particles represent distinct stylistic variations for the same input, suggesting that the model effectively generates multiple plausible perturbations simultaneously.

# 7. Other Works and Discussion

Our work is positioned at the intersection of sampling via gradient flow and robust optimization. There is a thread of works that considered the (D)RO problems via gradient descent-ascent dynamics (Wang & Chizat, 2022; Yu et al., 2022; García Trillos & García Trillos, 2024; Conger et al., 2023). They typically use proximal algorithms in the Wasserstein space *iteratively*, i.e., the Jordan-Kinderlehrer-Otto (JKO) scheme in PDE , which can be viewed as one-step discretization of the continuous time PDE $\partial_t \rho = \nabla \cdot (\rho \nabla \xi)$ for some $\xi$ . In this paper, we shift the perspective and view the inner maximization problem as a regularized variational problem that exactly corresponds to a sampling algorithm, without explicitly resorting to JKO iteratively. This allows our methodology to serve as a general-purpose sampler-based bi-level optimization framework (Nemirovski et al., 2009), enabling a direct approach based on optimizing measures.

This perspective possesses significant unifying power: it reveals that previous methods such as WRM (Sinha et al., 2018) are fundamentally rooted in ODE analysis, which can be seen as a special case of the S/PDE framework proposed here. Our framework is also orthogonal to map-based

approaches such as FlowDRO (Xu et al., 2024) and JKO-iterative analyses such as Zhu et al. (2025): we address DRO via a sampling methodology built on the equivalence between the inner problem and an entropy-regularized JKO step. This also allows optimizers to go beyond the standard (Wasserstein) gradient descent type of algorithm, inspiring novel approaches such as SVG and WFR gradient flows . Our theory treats general DRO by simply changing the corresponding gradient flow energy function $F$ in (9), opening the door for future research into diverse DRO ambiguity notions and gradient flow dissipation geometries.

A practical SGDA-style variant of our framework is also possible. Instead of running the inner sampler to convergence, one can reuse the perturbed samples from the previous iteration and take only a few inner update steps before each outer parameter update. A full theoretical analysis of this SGDA variant is left to future work.

## Acknowledgement.

We thank Jie Wang (Wang, 2025) for pointing out a flaw in our original proof of Theorem 5.5, which we have now fixed. We acknowledge the support from the Deutsche Forschungsgemeinschaft (DFG, German Research Foundation) as part of the priority programme "Theoretical Foundations of Deep Learning" (project number: 543963649).

## Impact Statement

This paper presents work whose goal is to advance the field of Machine Learning. There are many potential societal consequences of our work, none which we feel must be specifically highlighted here.

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

# A. Further Background

## A.1. Dual approach to Entropy-regularized Wasserstein DRO

The Sinkhorn DRO problem, as formulated by Wang et al. (2025), aims to find a decision that minimizes the worst-case expected loss over an ambiguity set defined by Sinkhorn divergence, which is identical to entropy-regularized Wasserstein distance in this problem. Given a loss function $\ell(\theta, z)$, a nominal distribution $\widehat{\rho_N}$, and a radius $r > 0$, the primal inner problem is:

$$\sup_{\rho \in \mathcal{B}_{r,\epsilon}(\widehat{\rho_N})} \mathbb{E}_{z \sim \rho}[\ell(\theta, z)], \tag{20}$$

where the ambiguity set is $\mathcal{B}_{r,\epsilon}(\widehat{\rho_N}) := \{\rho \in \mathcal{P}(\mathcal{Z}) : W_\epsilon^2(\widehat{\rho_N}, \rho) \leq r\}$, and the entropy-regularized Wasserstein distance,

$$W_\epsilon^2(\rho_1, \rho_2) = \inf_{\gamma \in \Gamma(\rho_1, \rho_2)} \{\mathbb{E}_{(x,y) \sim \gamma}[c(x, y)] + H(\gamma)\}.$$

A dual approach, established by Wang et al. (2025), reformulates the inner maximization problem. Using the strong duality of this problem and substituting (7) into this problem, the inner problem can be expressed as a minimization over a dual variable $\tau$:

$$\inf_{\tau \geq 0} \left\{ \frac{r}{2\tau} + \frac{\epsilon}{2\tau} \mathbb{E}_{x \sim \widehat{\rho_N}} \left[ \log \mathbb{E}_{y \sim \rho_{x,\epsilon}} \left[ \exp \left( \frac{2\tau \ell(\theta, y)}{\epsilon} \right) \right] \right] \right\}, \tag{21}$$

where $\rho_{x,\epsilon}(y) = \frac{\exp(\frac{-c(x,y)}{\epsilon})}{Z}$ is a kernel distribution centered at $x$, and $Z$ is a normalization constant.

To approximate the value and the gradient of the nested expectation in (21), Wang et al. (2025) employ a two-level sampling procedure for a given $\tau$: first, a sample $x$ is drawn from the nominal distribution $\widehat{\rho_N}$; second, a set of samples are drawn from the kernel distribution $\rho_{x,\epsilon}$ to approximate the inner expectation.

However, a major limitation of this dual method is that the nested expectation structure in the objective leads to a biased subgradient estimator when using standard Monte Carlo sampling, especially for small values of $\epsilon$. This bias can impede the convergence of the optimization procedure. This challenge motivates the exploration of alternative approaches, such as the gradient flow methods discussed in this paper, which can directly sample from the primal worst-case distribution to obtain a less biased estimate of the gradient.

*Remark* A.1. Consider the KL-DRO problem:

$$\min_\theta \sup_{\rho \ll \rho_0} \left\{ \mathbb{E}_{y \sim \rho}[\ell(\theta, y)] - \frac{1}{2\tau} \mathrm{KL}(\rho \| \rho_0) \right\} \tag{22}$$

Let us define a reference distribution $\rho_0(y)$ as a kernel density estimator of the empirical distribution $\widehat{\rho_N} = \frac{1}{N} \Sigma_{i=1}^N \delta(x_i)$: $\rho_0(y) = \frac{1}{N} \Sigma_{i=1}^N K_\epsilon(y, x_i)$, where $K_\epsilon(y, x_i) = \alpha_\epsilon \exp(-c(y, x_i)/\epsilon)$ and $\alpha_\epsilon = \mathbb{E}[\exp(-c(y, x_i)/\epsilon)]^{-1}$. With this choice, and by setting $\tau = \tau'/\epsilon$, the solution to the inner maximization problem is given by Hu & Hong (2013):

$$\rho^{\mathrm{KL}}(y) = \rho_0(y) \cdot \frac{\exp(2\tau' \ell(\theta, y)/\epsilon)}{\mathbb{E}_{y \sim \rho_0}[\exp(2\tau' \ell(\theta, y)/\epsilon)]}$$

$$= \frac{1}{N} \Sigma_{i=1}^N \alpha_\epsilon \beta \exp((2\tau' \ell(\theta, y) - c(y, x_i))/\epsilon),$$

where $\beta = \mathbb{E}_{y \sim \rho_0}[\exp(2\tau' \ell(\theta, y)/\epsilon)]^{-1}$. This resulting distribution bears a strong resemblance to the worst-case distribution for entropy-regularized DRO. However, they are not identical, as the normalization constants differ: $\alpha_\epsilon \beta \neq \alpha_x := \mathbb{E}[\exp((2\tau' \ell(\theta, y) - c(y, x))/\epsilon)]^{-1}$. In fact, the term $\alpha_\epsilon \beta \exp((2\tau' \ell(\theta, y) - c(y, x))/\epsilon)$ does not integrate to one and thus does not define a valid conditional probability measure. Actually, $\rho^{\mathrm{KL}}(y)$ can be interpreted as a weighted expectation of conditional distributions:

$$\rho^{\mathrm{KL}} = \mathbb{E}_{x \sim \widehat{\rho_N}} \left[ \frac{\alpha_\epsilon \beta}{\alpha_x} \rho_{Y|X=x} \right]$$

Consequently, the two-layer sampling procedure requires computing the corresponding weights, sampling from the resulting weighted empirical distribution, and subsequently sampling from the conditional distribution. However, the term $\alpha_x$ is intractable in practice. Therefore, unlike the entropy-regularized DRO case, the KL-DRO problem cannot be directly framed as a two-level sampling task.

### A.2. Hellinger-Kantorovich a.k.a. Wasserstein-Fisher-Rao Gradient Flows

Consider the WFR gradient system of the energy functional $F$, i.e., the triple $(\mathcal{P}, F, \mathsf{WFR})$, where $\mathsf{WFR}$ is the Wasserstein-Fisher-Rao metric, a.k.a. the *Hellinger-Kantorovich* metric restricted to the probability space. Specifically, we consider the HK/WFR gradient system associated with reaction-diffusion PDE:

$$\partial_t \rho = \alpha \operatorname{div}\left(\rho \nabla \frac{\delta F}{\delta \rho}\right) - \beta\, \rho\left(\frac{\delta F}{\delta \rho} - \int \frac{\delta F}{\delta \rho}\, \mathrm{d}\rho\right). \tag{23}$$

The hope of this gradient flow is that the added reaction term on the right-hand side will help the gradient flow to converge more rapidly. This can be formally seen by checking the time derivative of the energy functional $F$ along the gradient flow (13), we have

$$\partial_t F(\rho_t) \overset{\text{(EDB)}}{=} -\alpha \int \left|\nabla \log \frac{\rho}{\pi}(x)\right|^2 \mathrm{d}\rho(x) - \beta \int \left|\log \frac{\rho}{\pi}(x) - \mathrm{KL}(\rho|\pi)\right|^2 \mathrm{d}\rho(x)$$

where we used the Energy-Dissipation Balance (EDB, a.k.a. equality) of gradient flows. We observe the right-hand side is non-positive, hence its dissipation will be faster than the original pure Wasserstein gradient flow.

However, we must note that there is no global PL or strong convexity analogous to the case of the Wasserstein gradient flow, as discussed in the main text. A "warm-start" condition is necessary as it has been shown that global PL/gradient dominance cannot hold for the Fisher-Rao/(spherical)Hellinger gradient flows; see Mielke & Zhu (2025); Carrillo et al. (2026) for details. There exist fine-grained warm-start initialization conditions in the gradient flow literature; see Lu et al. (2023; 2019); Chen et al. (2023).

For example, Lu et al. (2023) showed that, after a warm-start initialization in the form of a density-ratio lower bound, it is possible to get an improved exponential convergence rate than the pure Wasserstein gradient flow as characterized in Lemma C.2. However, it is important to note that the specific warm-start initialization condition cannot be verified in our DRO problems. Nonetheless, in practice, we observe significant speedups using the WFR gradient flow as compared to the pure Wasserstein gradient flow.

## B. Further Methods

### B.1. Wasserstein Fisher-Rao Gradient Flow-based DRO

This subsection details the DRO method based on the Wasserstein Fisher-Rao gradient flow. Algorithm 4 provides a comprehensive summary of this procedure. Compared to the WGF-based method, this approach further refines the DRO framework by introducing birth-death mechanism.

The WFR gradient flow associated with the energy $F$ in (9) is governed by the reaction–diffusion PDE (13). Discretizing it via interacting particles, each particle $(y^i, w^i)$ evolves through two coupled updates: a *transport update* identical to ULA, and a *reaction update* on the weights:

$$y_{t+1}^i = y_t^i - \eta\, \nabla \widetilde{V}_{x,\tau}(y_t^i) + \sqrt{\eta\epsilon/\tau}\, \xi_t^i, \qquad \text{(transport)}$$

$$w_{t+1}^i = (w_t^i)^{1-\epsilon\eta_w/(2\tau)} \cdot \exp\left(-\eta_w\, \widetilde{V}_{x,\tau}(y_t^i)\right). \qquad \text{(reaction)}$$

The reaction term increases or decrease the weights of particles based on their energy, low-energy particles gain mass, while high-energy ones lose mass. Once a particle's weight drops below a threshold $w_{\min}$, it is reborn at the location of a randomly selected surviving particle, with the two particles sharing the original mass equally. This birth–death mechanism prevents the wasted computation of tracking particles stuck in local minima, which we observe empirically in Figure 10 of Appendix D.4.

The full algorithm is summarized in Algorithm 4.

### B.2. Stein Variational Gradient-based DRO

Stein Variational Gradient Descent (SVGD) is a variational inference algorithm that approximates a target probability density $p(x)$ by iteratively transporting a set of particles $\{x_i\}_{i=1}^n$ (Liu & Wang, 2016). The method is formulated as a functional

---

**Algorithm 4** Entropy-regularized Wasserstein DRO via WFR flow

---

1: **Input:** Empirical distribution $\widehat{\rho_N}$, constraint set $\Theta$, $\tau, \epsilon > 0$, stepsize sequence $\{r_s > 0\}_{s=0}^{S-1}$, inner stepsize $\eta$, weight stepsize $\eta_w$, weight threshold $w_{\min}$, inner iterations $T$, number of samples $m$.
2: **for** $s = 0, \ldots, S - 1$ **do**
3:     Sample $x^s \sim \widehat{\rho_N}$
4:     Initialize $y_0^{i,s} \leftarrow x^s, i = 1, \ldots, m$
5:     **for** $t = 0, \ldots, T - 1$ **do**
6:         Sample $\xi_t^{i,s} \sim \mathcal{N}(0, I)$
7:         $y_{t+1}^{i,s} \leftarrow y_t^{i,s} - \eta \nabla \widetilde{V}_{x^s, \tau}(y_t^{i,s}) + \sqrt{\eta \epsilon / \tau} \xi_t^{i,s}$
8:         $w_{t+1}^{i,s} \leftarrow (w_t^{i,s})^{1 - \epsilon \eta_w / (2\tau)} e^{-\eta_w \widetilde{V}_{x^s, \tau}(y_t^{i,s})}$
9:         Normalize weights: $w_{t+1}^{i,s} \leftarrow w_{t+1}^{i,s} / \sum_{j=1}^{m} w_{t+1}^{i,s}$
10:         **Particle death/rejuvenation**:
11:         **if** $w_{t+1}^{i,s} < w_{\min}$ **then**
12:             Uniformly select an index $i'$ from $\{j : j \neq i\}$.
13:             $y_{t+1}^{i,s} \leftarrow y_{t+1}^{i',s}, \; w_{t+1}^{i',s}, w_{t+1}^{i,s} \leftarrow (w_{t+1}^{i,s} + w_{t+1}^{i',s})/2.$
14:         **end if**
15:     **end for**
16:     $\theta^{s+1} \leftarrow \text{Proj}_\Theta(\theta^s - r_s \sum_{i=1}^{m} w_T^{i,s} \nabla_\theta \ell(\theta^s, y_T^{i,s}))$
17: **end for**
18: **return** $\theta^S$

---

gradient descent on the Kullback-Leibler (KL) divergence, $\text{KL}(q\|p)$, where $q$ represents the empirical distribution of the particles. The particle updates are governed by a velocity field $\phi(x_i)$ that corresponds to the direction of steepest descent. For an empirical measure of $n$ particles, this update is given by:

$$\phi(x_i) \propto \frac{1}{n} \sum_{j=1}^{n} \left[ k(x_j, x_i) \nabla_{x_j} \log p(x_j) + \nabla_{x_j} k(x_j, x_i) \right]$$

This update rule can be decomposed into two functional components: (i) a driving force, which is a kernel-smoothed average of the score function $\nabla \log p(x)$ that directs particles towards the modes of the target distribution, and (ii) a repulsive force, which arises from the kernel gradient $\nabla k(\cdot, \cdot)$ and ensures particle diversity to prevent mode collapse (Ba et al., 2021).

The iterative application of this update rule forms the basis of the SVGD algorithm. As detailed in (Liu & Wang, 2016), its computational complexity is dominated by the pairwise kernel computations, resulting in a cost of $O(m^2 d)$ per iteration for $m$ particles in $d$ dimensions. Regarding convergence, theoretical guarantees have been established under certain assumptions. As shown by (Salim et al., 2022), for target distributions that satisfy Talagrand's T1 inequality, a finite-iteration complexity bound is derived. A finite-particle convergence analysis is also provided in (Shi & Mackey, 2023).

By applying SVGD to sample from the worst-case distribution, we arrive at Algorithm 5.

However, the empirical performance of Algorithm 5 is suboptimal. We observe that the convergence is sensitive to the initialization of the particles. Furthermore, when the number of particles $m$ is small, the magnitude of the driving force to dominate the repulsive force, thereby diminishing the variance. In this regime, the SVGD algorithm degenerates into a simple gradient descent method, losing its sampling capabilities. This behavior is demonstrated empirically in Appendix D.4.

### B.3. Restricted Gaussian Oracle-based DRO

In this section, we explore an alternative approach for sampling from the worst-case distribution based on Proximal Sampler. The *Proximal Sampler* (Lee et al., 2021) was introduced to unbiasedly sample from Gibbs distributions of the form $\nu(x) \propto \exp(-f(x))$. A single iteration of the proximal sampler consists of the following two steps:

1. (**Forward step**): Sample $y_k \mid x_k \sim \nu_\tau^{Y|X}(\cdot \mid x_k) = \mathcal{N}(x_k, \tau I)$. This results in a new iterate $y_k \sim \rho_k^Y$, where $\rho_k^Y = \rho_k^X * \mathcal{N}(0, \tau I)$.

2. (**Backward step**): Sample $x_{k+1} \mid y_k \sim \nu_\tau^{X|Y}(\cdot \mid y_k)$. This gives the next iterate $x_{k+1} \sim \rho_{k+1}^X$.

---

**Algorithm 5** Sinkhorn DRO via SVGD

---

1: **Input:** Empirical distribution $\widehat{\rho_N}$, constraint set $\Theta$, $\tau, \epsilon > 0$, stepsize sequence $\{r_s > 0\}_{s=0}^{S-1}$, inner stepsize $\eta$, inner iterations $T$, initial deviation $\sigma$, number of samples $m$.
2: **for** $s = 0, \ldots, S - 1$ **do**
3:     Sample $x^s \sim \widehat{\rho_N}$
4:     Sample $\xi^{s,i} \sim \mathcal{N}(0, \sigma I)$, $i = 1, \ldots, m$
5:     Initialize $y_0^{s,i} \leftarrow x^s + \xi^{s,i}$
6:     **for** $t = 0, \ldots, T - 1$ **do**
7:         $\phi_t^{s,i} \leftarrow \frac{1}{m} \sum_{j=1}^m \left[ -k(y_t^{s,i}, y_t^{s,j}) \cdot \frac{2\tau}{\epsilon} \nabla \widetilde{V}_{x^s, \tau}(y_t^{s,j}) + \nabla_2 k(y_t^{s,i}, y_t^{s,j}) \right]$
8:         $y_{t+1}^{s,i} \leftarrow y_t^{s,i} + \eta \phi_t^{s,i}$
9:     **end for**
10:    $\theta^{s+1} \leftarrow \text{Proj}_\Theta(\theta^s - r_s \sum_{i=1}^m \frac{1}{m} \nabla_\theta \ell(\theta^s, y_T^{s,i}))$
11: **end for**
12: **return** $\theta^S$

---

As shown in recent work (Chen et al., 2022), this forward-backward procedure is equivalent to an entropy-regularized Jordan-Kinderlehrer-Otto (JKO) scheme :

$$\rho_k^Y = \arg \min_{\mu \in P_2(\mathbb{R}^d)} \left\{ \frac{1}{2\tau} W_{2\tau}^2(\rho_k^X, \mu) \right\}, \tag{24}$$

$$\rho_{k+1}^X = \arg \min_{\mu \in P_2(\mathbb{R}^d)} \left\{ \int f \, d\mu + \frac{1}{2\tau} W_{2\tau}^2(\rho_k^Y, \mu) \right\}, \tag{25}$$

This connection to the entropy-regularized JKO scheme provides a new perspective to our problem. By reformulating our original Lagrangian problem ((1)) as a minimization and specializing the cost to the squared Euclidean distance, we obtain:

$$\Phi(\theta) = \min_{\rho \in \mathcal{P}} \{ \mathbb{E}_{y \sim \rho}[-\ell(\theta, y)] + \frac{1}{2\tau} W_\epsilon^2(\rho, \widehat{\rho_N}) \}, \tag{26}$$

which precisely matches the form of the problem (25) solved by backward step. This structural equivalence suggests that methods developed for proximal sampler can be directly applied to solve the entropy-regularized DRO problem. Specifically, we can employ the *Restricted Gaussian Oracle (RGO)* from Lee et al. (2021) to solve this. The sampling process for a given $x \sim \widehat{\rho_N}$ involves two steps:

1. Sample $x$ from $\widehat{\rho_N}$, and compute the minimizer $y_{x,\tau}^* = \arg \min_{y \in \mathbb{R}^d} \{ \widetilde{V}_{x,\tau}(y) \}$.

2. Repeat until acceptance: draw a sample $Z \sim \mathcal{N}\left(y_{x,\tau}^*, \frac{\epsilon}{2(1-L\tau)} I\right)$ and accept it with probability

$$\exp\left( -\widetilde{V}_{x,\tau}(Z) + \widetilde{V}_{x,\tau}(y_{x,\tau}^*) + \frac{2(1-L\tau)}{\epsilon} \|Z - y_{x,\tau}^*\|^2 \right).$$

The validity of the RGO sampler requires the loss function $\ell$ to be $L$-smooth, and the penalty parameter $\tau$ must satisfy $\tau < 1/L$. This condition ensures that the auxiliary function $\widetilde{V}_{x,\tau}(y)$ is $(\frac{2(1-L\tau)}{\epsilon})$-strongly convex, which is crucial for rejection sampling. Applying RGO to the entropy-regularized DRO problem necessitates that the loss function is $L$-smooth and requires solving a convex optimization subproblem in each step. These requirements on the loss function and the computational structure are analogous to the WRM algorithm presented in (Sinha et al., 2018) for the specific case of a squared Euclidean cost. However, a key difference exists: the WRM algorithm transports each data point to a single worst-case point, whereas in entropy-regularized DRO, each data point induces a full continuous distribution of worst-case scenarios.

*Remark* B.1. Although WRM(Sinha et al., 2018) assumes an $L$-smooth loss, its algorithm is "parameter-free" – it never needs to know $L$ and works whenever the inner optimization problem admits a closed-form solution. In contrast, Algorithm 6 requires explicit knowledge of $L$ and cannot be applied if the loss fails to be $L$-smooth. In practice, we may assume $\tau$ is sufficiently small such that $\tau < 1/L$, and $\ell$ is $L$-smooth for all $y$.

---

**Algorithm 6** Sinkhorn DRO via RGO

---

1: **Input:** Empirical distribution $\widehat{\rho_N}$, constraint sets $\Theta$, $\tau, \epsilon > 0$, stepsize sequence $\{r_s > 0\}_{s=0}^{S-1}$
2: **for** $s = 0, \ldots, S - 1$ **do**
3:     Sample $x^s \sim \widehat{\rho_N}$ and find an $\eta$-approximate minimizer $\hat{y}^s$ of $\widetilde{V}_{\tau, x^s}(y)$
4:     Generate samples $\{y^{s,i}\}_{i=1}^m$ via rejection sampling
5:     $\theta^{s+1} \leftarrow \mathrm{Proj}_\Theta(\theta^s - \frac{r_s}{m} \sum_{i=1}^m \nabla_\theta \ell(\theta^s, y^{s,i}))$
6: **end for**
7: **return** $\theta^S$

---

# C. Proofs of Theoretical Results

## C.1. Proof of Proposition 3.1

*Proof of Proposition 3.1.* We rewrite the problem using the definition of the entropy-regularized OT formulation:

$$\min_\rho \min_\Pi \left\{ \int V \, \mathrm{d}\rho + \frac{1}{2\tau} \int c(x, y) \, \mathrm{d}\Pi + \frac{\epsilon}{2\tau} \int \log \Pi \, \mathrm{d}\Pi \right\} \tag{27}$$

subject to the margianl constraints $\int \Pi \, \mathrm{d}x = \rho$ and $\int \Pi \, \mathrm{d}y = \rho_0$. Here, $\rho_0$ can be set to empirical distribution $\widehat{\rho_N}$ as in the data-driven DRO problem. Using the marginal constraints, we can then write the first linear term as $\int \int V(y) \, \mathrm{d}\Pi(x, y)$ and eliminate the variable $\rho$, i.e., one end is unconstrained – hence it's the half bridge problem:

$$\min_\Pi \left\{ \int V \, \mathrm{d}\Pi + \frac{1}{2\tau} \int c(x, y) \, \mathrm{d}\Pi + \frac{\epsilon}{2\tau} \int \log \Pi \, \mathrm{d}\Pi \,\middle|\, \int \Pi \, \mathrm{d}y = \rho_0 \right\}. \tag{28}$$

After straightforward linear optimization, we obtain the optimal solution to the variational problem (5) as

$$\rho^*(y) = \mathbb{E}_{x \sim \rho_0} \left[ \frac{1}{Z_x} \exp\left[ -\frac{2\tau V(y) + c(y, x)}{\epsilon} \right] \right], \quad Z_x = \int \exp\left[ -\frac{2\tau V(y) + c(y, x)}{\epsilon} \right] \mathrm{d}y. \tag{29}$$

Using elementary manipulations, the variational problem (5) can also be written down as the minimization of the expected KL divergence:

$$\min_{\rho_{Y|X}} \mathbb{E}_{x \sim \rho_0} \mathrm{KL}\left( \rho_{Y|X=x}(y) \,\middle|\, \frac{1}{Z_x} \exp\left[ -\frac{2\tau V(y) + c(y, x)}{\epsilon} \right] \right). \tag{30}$$

$\square$

## C.2. Proof of Proposition 4.2

First, we recall some standard results.

**Lemma C.1.** *Suppose the first condition in Definition 4.1 holds for some $\lambda > 0$. Then, $F$ is $\lambda$-displacement convex.*

**Lemma C.2** (WGF gradient dominance/PL)**.** *Let $\rho_t$ be a solution to the Wasserstein gradient flow of the KL divergence energy functional $\mathrm{KL}(\rho | \exp(-\widetilde{V}_{x,\tau}))$. Then,*

$$\widetilde{V} \text{ is } \lambda\text{-displacement convex or } \lambda\text{-PL} \implies \mathrm{KL}(\rho_t | \exp(-\widetilde{V}_{x,\tau})) \le e^{-2\lambda t} \, \mathrm{KL}(\rho_0 | \exp(-\widetilde{V}_{x,\tau})).$$

**Lemma C.3** (Talagrand's inequality)**.** *Under the same assumptions as in Lemma C.2, we have*

$$W_2^2(\rho_t, \exp(-\widetilde{V}_{x,\tau})) \le \sqrt{\frac{2}{\lambda} \mathrm{KL}(\rho_t | \exp(-\widetilde{V}_{x,\tau}))}$$

Recall also that $W_1(\rho_t, \pi_{Y|X=x}) \le W_2(\rho_t, \pi_{Y|X=x})$.

**From conditional to marginal bound** We have the following relation between the marginal and conditional KL divergences. Suppose $\rho_X$ is the marginal distribution of $X$. Recall the relation $\pi_Y = \int_x \rho_{Y|X=x}^* \, \mathrm{d}\rho_X(x)$. Then,

$$\mathrm{KL}(\rho_Y | \pi_Y) \le \mathbb{E}_{X \sim \rho_X} \left[ \mathrm{KL}\left( \rho_{Y|X} \middle| \pi_{Y|X} \right) \right].$$

Alternatively, the Wasserstein relation gives

$$W_p(\rho_Y, \pi_Y) \leq \mathbb{E}_{X \sim \rho_X} \left[ W_p(\rho_{Y|X}(\cdot|X), \pi_{Y|X}(\cdot|X)) \right].$$

**Marginal distribution error control** Suppose we have the samples $y_t^i \sim \rho_{t,Y}$, Our goal is to use the empirical measure $\widehat{\rho_{t,Y}} = \frac{1}{N} \sum_{i=1}^{N} \delta_{y_t^i}$ to approximate $\pi_Y$.

Taking expectations and applying the triangle inequality, we have

$$\mathbb{E}[W_1(\widehat{\rho_{t,Y}}, \pi_Y)] \leq \mathbb{E}[W_1(\widehat{\rho_{t,Y}}, \rho_{t,Y})] + W_1(\rho_{t,Y}, \pi_Y), \tag{31}$$

where the first term is from particle approximation, which vanishes as $N \to \infty$. And the second term is the optimization error along the gradient flow. By Lemma C.2 and Lemma C.3, together with the conditional-to-marginal Wasserstein bound,

$$W_1(\rho_{t,Y}, \pi_Y) \leq \mathbb{E}_{X \sim \rho_X} \left[ W_2(\rho_{t,Y|X}, \pi_{Y|X}) \right] \lesssim \frac{\mathrm{e}^{-\lambda t}}{\sqrt{\lambda}}. \tag{32}$$

Therefore, to reach a $\delta_{\text{sample}}$-solution in expectation, we need to run the gradient flow (i.e. simulating the PDE/SDE) for time

$$t \gtrsim \frac{1}{\lambda} \log \frac{1}{\sqrt{\lambda} \delta_{\text{sample}}}. \tag{33}$$

We will later show the discrete-time gradient descent version of this result.

We emphasize that the continuous-time gradient flow result is important not only for understanding the limit of discrete-time algorithms, but also for serving as a guide for designing new discrete-time algorithms. Let us get a quick insight from the above estimate: suppose we discretized algorithm with discretization stepsize $\eta$ for in total $T$ steps. Ignoring the discretization error, we have $t = T\eta$ and the (ideal) number of iterations needed to reach a $\delta_{\text{sample}}$-solution of the inner maximization problem (14) is $T \gtrsim \frac{1}{\eta\lambda} \log \frac{1}{\sqrt{\lambda}\delta_{\text{sample}}}$.

**DRO gradient oracle error control via gradient flow** Let us now show how to convert the error estimate in distributions to the error estimate in the gradient oracle, needed for the optimization analysis later. If $\mathbb{E}[W_1(\widehat{\rho_{t,Y}}, \pi_Y)] \leq \delta_{\text{sample}}$ for some $\delta_{\text{sample}} > 0$ (as controlled by (32)) and $g$ is $L$-Lipschitz (w.r.t. Euclidean norm). Then, by Kantorovich–Rubinstein duality (Kantorovich, 2006), we obtain:

$$\mathbb{E}\left[ \left| \left| \int g \, d(\widehat{\rho_{t,Y}} - \pi_Y) \right| \right| \right] \leq L \cdot \mathbb{E}[W_1(\widehat{\rho_{t,Y}}, \pi_Y)] \leq L\delta_{\text{sample}}. \tag{34}$$

In the optimization analysis, we can later take $g$ to be the gradient oracle of the DRO loss, $g := \nabla_\theta \ell(\theta, \cdot)$, to control the gradient oracle error. Recall that $\widehat{\rho_{t,Y}} = \frac{1}{N} \sum_{i=1}^{N} \delta_{y_t^i}$ is the particle approximation obtained by running the gradient flow sampler for $t$ steps, and $\pi_Y$ is the target marginal distribution, which is the worst-case distribution of DRO inner maximization problem (14). Then, (34) results in the following bound on the gradient oracle error:

$$\mathbb{E}\left[ \left| \left| \frac{1}{N} \sum_{i=1}^{N} \nabla_\theta \ell(\theta, y_t^i) - \nabla_\theta \underbrace{\max_{\rho \in \mathcal{P}} \left( \int \ell(\theta, z) \, d\rho(z) - \frac{1}{2\tau} W_\epsilon^2(\rho, \widehat{\rho_N}) \right)}_{\text{DRO objective}} \right| \right| \right]$$

$$= \mathbb{E}\left[ \left| \left| \frac{1}{N} \sum_{i=1}^{N} \nabla_\theta \ell(\theta, y_t^i) - \mathbb{E}_{y \sim \pi_Y} \nabla_\theta \ell(\theta, y) \right| \right| \right] \overset{(34)}{\leq} L\delta_{\text{sample}} \tag{35}$$

Replacing $L\delta_{\text{sample}}$ by $\epsilon_{\text{grad}}$, we obtain the desired bound on the gradient oracle error as in Proposition 4.2.

### C.3. Proof of Theorem 5.2 (Outer Loop Convergence)

We provide a detailed proof for the convergence of the outer loop, which follows the standard analysis for non-convex Stochastic Gradient Descent with a persistent bias.

**Lemma C.4** (Bias of the Stochastic Gradient). *Under Assumptions 5.1.2, 5.1.3, the bias of the stochastic gradient estimator, defined as $B(\theta) := \mathbb{E}[\widehat{g}] - \nabla\Phi(\theta)$, is bounded as:*

$$\|B(\theta)\| \leq L_f \cdot \delta_{sample}.$$

*Proof.* By definition, the bias is

$$B(\theta) = \mathbb{E}_{x\sim\widehat{\rho_N}}\left[\mathbb{E}_{y\sim\widehat{\rho}_{Y|X=x}}[\nabla_\theta\ell(\theta,y)]\right] - \mathbb{E}_{x\sim\widehat{\rho_N}}\left[\mathbb{E}_{y\sim\pi_{Y|X=x}}[\nabla_\theta\ell(\theta,y)]\right]$$

$$= \mathbb{E}_{x\sim\widehat{\rho_N}}\left[\mathbb{E}_{y\sim\widehat{\rho}_{Y|X=x}}[\nabla_\theta\ell(\theta,y)] - \mathbb{E}_{y\sim\pi_{Y|X=x}}[\nabla_\theta\ell(\theta,y)]\right].$$

Taking norms and using Jensen's inequality:

$$\|B(\theta)\| \leq \mathbb{E}_{x\sim\widehat{\rho_N}}\left[\|\mathbb{E}_{y\sim\widehat{\rho}_{Y|X=x}}[\nabla_\theta\ell(\theta,y)] - \mathbb{E}_{y\sim\pi_{Y|X=x}}[\nabla_\theta\ell(\theta,y)]\|\right].$$

By Assumption 5.1.2, the map $y \mapsto \nabla_\theta\ell(\theta,y)$ is $L_f$-Lipschitz in $y$. Hence by Kantorovich–Rubinstein duality, for any two probability measures $\rho_1, \rho_2$ with finite first moment,

$$\left\|\mathbb{E}_{y\sim\rho_1}[\nabla_\theta\ell(\theta,y)] - \mathbb{E}_{y\sim\rho_2}[\nabla_\theta\ell(\theta,y)]\right\| \leq L_f \cdot W_1(\rho_1,\rho_2).$$

Applying this and using $W_1 \leq W_2$, we obtain

$$\begin{aligned}
\|B(\theta)\| &\leq \mathbb{E}_{x\sim\widehat{\rho_N}}\left[L_f \cdot W_1(\widehat{\rho}_{Y|X=x}, \pi_{Y|X=x})\right] \\
&\leq L_f \cdot \mathbb{E}_{x\sim\widehat{\rho_N}}\left[W_2(\widehat{\rho}_{Y|X=x}, \pi_{Y|X=x})\right] \quad \text{(since } W_1 \leq W_2) \\
&\leq L_f \cdot \delta_{\text{sample}}.
\end{aligned}$$

$\square$

**Proof of Theorem 5.2.** From the $L_\Phi$-smoothness of $\Phi$ (Assumption 5.1.1), we have the standard descent lemma. For simplicity, we assume $\Theta = \mathbb{R}^d$ and a constant stepsize $r_s = r$.

$$\begin{aligned}
\Phi(\theta^{s+1}) &\leq \Phi(\theta^s) + \langle\nabla\Phi(\theta^s), \theta^{s+1} - \theta^s\rangle + \frac{L_\Phi}{2}\|\theta^{s+1} - \theta^s\|^2 \\
&= \Phi(\theta^s) - r\langle\nabla\Phi(\theta^s), \widehat{g}^s\rangle + \frac{L_\Phi r^2}{2}\|\widehat{g}^s\|^2.
\end{aligned}$$

Taking expectation conditioned on the filtration $\mathcal{F}_s$:

$$\begin{aligned}
\mathbb{E}[\Phi(\theta^{s+1})|\mathcal{F}_s] &\leq \Phi(\theta^s) - r\langle\nabla\Phi(\theta^s), \mathbb{E}[\widehat{g}^s|\mathcal{F}_s]\rangle + \frac{L_\Phi r^2}{2}\mathbb{E}[\|\widehat{g}^s\|^2|\mathcal{F}_s] \\
&= \Phi(\theta^s) - r\langle\nabla\Phi(\theta^s), \nabla\Phi(\theta^s) + B(\theta^s)\rangle + \frac{L_\Phi r^2}{2}\left(\|\mathbb{E}[\widehat{g}^s|\mathcal{F}_s]\|^2 + \text{Var}(\widehat{g}^s|\mathcal{F}_s)\right).
\end{aligned}$$

Using the variance bound and the bias definition:

$$\mathbb{E}[\Phi(\theta^{s+1})|\mathcal{F}_s] \leq \Phi(\theta^s) - r\|\nabla\Phi(\theta^s)\|^2 - r\langle\nabla\Phi(\theta^s), B(\theta^s)\rangle + \frac{L_\Phi r^2}{2}\left(\|\nabla\Phi(\theta^s) + B(\theta^s)\|^2 + \sigma^2\right).$$

Applying Young's inequality to the inner product term:

$$-r\langle\nabla\Phi(\theta^s), B(\theta^s)\rangle \leq \frac{r}{2}\|\nabla\Phi(\theta^s)\|^2 + \frac{r}{2}\|B(\theta^s)\|^2.$$

Substituting this in and simplifying with $\|a + b\|^2 \leq 2\|a\|^2 + 2\|b\|^2$:

$$\mathbb{E}[\Phi(\theta^{s+1})|\mathcal{F}_s] \leq \Phi(\theta^s) - \frac{r}{2}\|\nabla\Phi(\theta^s)\|^2 + \frac{r}{2}\|B(\theta^s)\|^2 + L_\Phi r^2(\|\nabla\Phi(\theta^s)\|^2 + \|B(\theta^s)\|^2) + \frac{L_\Phi r^2 \sigma^2}{2}.$$

Rearranging terms to isolate $\|\nabla\Phi(\theta^s)\|^2$:

$$r\left(\frac{1}{2} - L_\Phi r\right)\|\nabla\Phi(\theta^s)\|^2 \leq \Phi(\theta^s) - \mathbb{E}[\Phi(\theta^{s+1})|\mathcal{F}_s] + \left(\frac{r}{2} + L_\Phi r^2\right)\|B(\theta^s)\|^2 + \frac{L_\Phi r^2 \sigma^2}{2}.$$

Choosing a stepsize $r \leq \frac{1}{4L_\Phi}$, we have $(\frac{1}{2} - L_\Phi r) \geq \frac{1}{4}$ and $(\frac{r}{2} + L_\Phi r^2) \leq r$. This gives:

$$\frac{r}{4}\|\nabla\Phi(\theta^s)\|^2 \leq \Phi(\theta^s) - \mathbb{E}[\Phi(\theta^{s+1})|\mathcal{F}_s] + r\|B(\theta^s)\|^2 + \frac{L_\Phi r^2 \sigma^2}{2}.$$

Taking total expectation and using Lemma C.4, $\|B(\theta^s)\|^2 \leq (L_f \delta_{\text{sample}})^2$:

$$\frac{r}{4}\mathbb{E}[\|\nabla\Phi(\theta^s)\|^2] \leq \mathbb{E}[\Phi(\theta^s)] - \mathbb{E}[\Phi(\theta^{s+1})] + r(L_f \delta_{\text{sample}})^2 + \frac{L_\Phi r^2 \sigma^2}{2}.$$

Summing from $s = 0$ to $S - 1$ yields a telescoping sum:

$$\frac{r}{4}\sum_{s=0}^{S-1}\mathbb{E}[\|\nabla\Phi(\theta^s)\|^2] \leq \Phi(\theta^0) - \mathbb{E}[\Phi(\theta^S)] + Sr(L_f \delta_{\text{sample}})^2 + \frac{SL_\Phi r^2 \sigma^2}{2}.$$

Dividing by $\frac{rS}{4}$ and using $\mathbb{E}[\Phi(\theta^S)] \geq \Phi_{\text{inf}}$:

$$\frac{1}{S}\sum_{s=0}^{S-1}\mathbb{E}[\|\nabla\Phi(\theta^s)|^2] \leq \frac{4(\Phi(\theta^0) - \Phi_{\text{inf}})}{rS} + 4(L_f \delta_{\text{sample}})^2 + 2L_\Phi r \sigma^2.$$

Therefore, when $\delta_{\text{sample}}$ is controlled such that $\delta_{\text{sample}} = O(\epsilon_{\text{opt}}/L_f)$, and step size $r = \sqrt{\frac{1}{SL_\Phi \sigma^2}}$, and the iteration $S = O(1/\epsilon_{\text{opt}}^4)$, $\frac{1}{S}\sum_{s=0}^{S-1}\mathbb{E}[\|\nabla\Phi(\theta^s)|^2] \leq O(\epsilon_{\text{opt}}^2)$.

## C.4. Derivation of ULA Sampler Complexity (for Theorem 5.5)

The complexity of the ULA sampler depends on the geometric properties of the inner target distribution $\pi_{Y|X=x}$. Under Assumption 5.3, standard results on ULA convergence (Vempala & Wibisono, 2019) state that after $T$ steps, the KL-divergence to the target is bounded. To achieve a final KL-divergence of $\delta_{KL}$, the number of iterations required is $T = O\left(\frac{L_U^2 d}{\lambda_U^2 \delta_{KL}} \log \frac{1}{\delta_{KL}}\right) = \tilde{O}\left(\frac{L_U^2 d}{\lambda_U^2 \delta_{KL}}\right)$.

Our goal is to connect the required outer-loop sampling accuracy $\delta_{\text{sample}}$ (in $W_2$ distance) to the required inner-loop KL-divergence accuracy $\delta_{KL}$. Under the LSI assumption, Talagrand's inequality gives the relationship:

$$W_2^2(\widehat{\rho}_{Y|X=x}, \pi_{Y|X=x}) \leq \frac{2}{\lambda_U}\text{KL}(\widehat{\rho}_{Y|X=x}||\pi_{Y|X=x})$$

From Theorem 5.2, we require $W_2(\widehat{\rho}_{Y|X=x}, \pi_{Y|X=x}) \leq \delta_{\text{sample}} = O(\epsilon_{\text{opt}}/L_f)$. This implies we need to achieve a KL-divergence of:

$$\delta_{KL} \leq \frac{\lambda_U}{2}W_2^2(\widehat{\rho}_{Y|X=x}, \pi_{Y|X=x}) = O\left(\lambda_U \frac{\epsilon_{\text{opt}}^2}{L_f^2}\right).$$

Substituting this required $\delta_{KL}$ into the ULA iteration complexity gives the number of inner loop steps:

$$T_{ULA} = \tilde{O}\left(\frac{L_U^2 d}{\lambda_U^2 \cdot \lambda_U \frac{\epsilon_{\text{opt}}^2}{L_f^2}}\right) = \tilde{O}\left(\frac{L_U^2 L_f^2 d}{\lambda_U^3 \epsilon_{\text{opt}}^2}\right).$$

The total complexity is the product of outer iterations $S = O(1/\epsilon_{\text{opt}}^4)$, inner iterations $T_{ULA}$, and the cost per inner gradient step $C_{\nabla_z} = O(d)$.

$$\text{Complexity} = S \times T \times C_{\nabla_z}$$

$$= O\left(\frac{1}{\epsilon_{\text{opt}}^4}\right) \times \tilde{O}\left(\frac{L_U^2 L_f^2 d}{\lambda_U^3 \epsilon_{\text{opt}}^2}\right) \times O(d)$$

$$= \tilde{O}\left(\frac{L_U^2 L_f^2 d^2}{\lambda_U^3 \epsilon_{\text{opt}}^6}\right).$$

Note that we absorb constants like $L_\Phi$ into the $\tilde{O}(\cdot)$ notation. This completes the derivation for Theorem 5.5.

### C.5. Complexity of RGO

We analyze the computational complexity of RGO method under specific regularity conditions on the loss function.

**Theorem C.5** (Complexity of RGO Sampling). *Let the loss function $\ell : \mathbb{R}^d \to \mathbb{R}$ be $L$-smooth. For parameter $\tau = \frac{1}{Ld}$ and a target distribution $\pi_{Y|X=x}$, through the rejection sampling procedure described in Appendix B.3, we can draw a sample from a distribution $\tilde{\rho}_{Y|X=x}$ such that the Kullback-Leibler (KL) divergence satisfies $\text{KL}(\tilde{\rho}_{Y|X=x} \| \pi_{Y|X=x}) < \delta$ with iteration complexity (where $\epsilon$ is the entropy regularization parameter from (1))*

$$O\left(\log\left(\frac{1}{\epsilon\delta}\right)\right).$$

*Proof.* Under the assumption that $\ell$ is $L$-smooth and $\tau < \frac{1}{L}$, $\widetilde{V}_{x,\tau}(y)$, is $\frac{2-2\tau L}{\epsilon}$-strongly convex and $\frac{2+2\tau L}{\epsilon}$-smooth.

Let the minimizer of $\widetilde{V}_{x,\tau}(y)$ as $y_{x,\tau}^*$. To find the minimum of $\widetilde{V}_{x,\tau}(y)$, we can apply gradient descent. The number of iterations required to achieve a $\delta'$-approximate optimal solution $\tilde{y}$ is determined by the condition number $\kappa = \frac{1+\tau L}{1-\tau L}$. Specifically, the iteration complexity is:

$$O\left(\kappa \log\left(\frac{1}{\delta'}\right)\right) = O\left(\frac{1+\tau L}{1-\tau L} \log\left(\frac{1}{\delta'}\right)\right)$$

**2. Distributional Proximity.** Let the distribution obtained via rejection sampling with a $\delta'$-approximate optimal solution for the proposal be denoted by $\tilde{\rho}_{Y|X=x}$. The procedure defines the proposal distributions $p(y)$ (using the true minimizer $y_{x,\tau}^*$) and $\tilde{p}(y)$ (using the approximate minimizer $\tilde{y}$) as Gaussian distributions:

$$p(y) = \mathcal{N}\left(y \mid y_{x,\tau}^*, \frac{\epsilon}{2-2\tau L} I\right), \quad \tilde{p}(y) = \mathcal{N}\left(y \mid \tilde{y}, \frac{\epsilon}{2-2\tau L} I\right)$$

The corresponding acceptance probabilities $a(y)$ and $\tilde{a}(y)$ are given by:

$$a(y) = \exp\left(-\widetilde{V}_{x,\tau}(y) + \widetilde{V}_{x,\tau}(y_{x,\tau}^*) + \frac{1-\tau L}{\epsilon}\|y - y_{x,\tau}^*\|^2\right)$$

$$\tilde{a}(y) = \min\left\{\exp\left(-\widetilde{V}_{x,\tau}(y) + \widetilde{V}_{x,\tau}(\tilde{y}) + \frac{1-\tau L}{\epsilon}\|y - \tilde{y}\|^2\right), 1\right\}$$

The resulting KL divergence from the distribution obtained with the true minimizer, $\rho_{\theta,x}$, is

$$\text{KL}(\tilde{\rho}_{Y|X=x}\|\pi_{Y|X=x}) = \int \frac{\tilde{a}(y)\tilde{p}(y)}{\int \tilde{a}(y)\tilde{p}(y)} \log\left(\frac{\frac{\tilde{a}(y)\tilde{p}(y)}{\int \tilde{a}(y)\tilde{p}(y)}}{\frac{a(y)p(y)}{\int a(y)p(y)}}\right) dy \tag{36}$$

$$= \int \frac{\tilde{a}(y)\tilde{p}(y)}{\int \tilde{a}(y)\tilde{p}(y)} \left(\log\left(\frac{\tilde{a}(y)\tilde{p}(y)}{a(y)p(y)}\right) + \log\left(\frac{\int a(y)p(y)}{\int \tilde{a}(y)\tilde{p}(y)}\right)\right) dy \tag{37}$$

$$\leq \int \frac{\tilde{a}(y)\tilde{p}(y)}{\int \tilde{a}(y)\tilde{p}(y)} \cdot 2\left(\widetilde{V}_{x,\tau}(\tilde{y}) - \widetilde{V}_{x,\tau}(y_{x,\tau}^*)\right) dy \tag{38}$$

$$\leq \int \frac{\tilde{a}(y)\tilde{p}(y)}{\int \tilde{a}(y)\tilde{p}(y)} \cdot \frac{4+4\tau L}{\epsilon}\delta' dy = \frac{4+4\tau L}{\epsilon}\delta' \tag{39}$$

To ensure the final KL divergence is less than $\delta$, we must set the optimization accuracy to $\delta' < \frac{\epsilon}{4+4\tau L}\delta$. This implies an optimization complexity of $O\left(\frac{1+\tau L}{1-\tau L}\log\left(\frac{4+4\tau L}{\epsilon\delta}\right)\right)$.

**3. Rejection Sampling Efficiency.** The expected number of iterations until acceptance is at most $\left(\frac{1+\tau L}{1-\tau L}\right)^{d/2}$ (Chewi et al., 2022). Note that if we choose $\tau = \frac{1}{Ld}$, then the expected number of iterations until acceptance is at most $\left(\frac{1+\tau L}{1-\tau L}\right)^{d/2} = \left(\frac{1+1/d}{1-1/d}\right)^{d/2} = O(1)$. As $d \to \infty$, this expression converges to $e$. Thus, for this choice of $\tau$, the expected number of rejection sampling trials is $O(1)$.

**4. Total Complexity.** The total complexity is the product of the optimization complexity and the expected number of rejection sampling iterations. With $\tau = \frac{1}{Ld}$, the condition number becomes $\kappa = \frac{d+1}{d-1}$. More precisely, the complexity is:

$$O\left(\left(\frac{d+1}{d-1}\right)^{\frac{d}{2}+1} \times \log\left(\frac{4d+4}{d\epsilon\delta}\right)\right) = O(\log\left(\frac{1}{\epsilon\delta}\right))$$

This completes the proof. $\qquad\qquad\qquad\qquad\qquad\qquad\qquad\qquad\qquad\qquad\qquad\qquad\qquad\qquad\square$

*Remark* C.6 (Practical Limitations). The theoretical complexity presented in Theorem C.5 is highly compelling when compared to alternative sampling methodologies. However, its applicability is constrained by stringent underlying assumptions.

1. **Smoothness Requirement:** The analysis presupposes that the loss function $\ell$ is $L$-smooth. In many practical applications, loss functions are non-smooth or exhibit a very large smoothness constant $L$.

2. **Parameter Dependency:** The optimal setting for the parameter, $\tau = \frac{1}{Ld}$, is inversely proportional to both the smoothness constant $L$ and the dimension $d$. If $L$ is large, the resulting $\tau$ will be small. This makes the worst-case distribution very close to the original distribution, thereby limiting the robustness conferred by the method.

Consequently, while the theoretical result is intriguing, the method's performance in empirical settings is often suboptimal due to the difficulty in satisfying these idealized conditions.

# D. Additional Experiments

### D.1. Ablation Study for Human Face Classification

In this section, we evaluate the sensitivity of the proposed WFR gradient flow sampler to its key hyperparameters. To ensure a controlled comparison, we vary one parameter at a time while keeping the others fixed at their baseline values. The baseline configuration is defined as follows: regularization $\tau = 5$, entropy regularization $\epsilon = 0.1$, learning rate $\eta = 0.01$, number of particles $m = 32$, and total iterations $T = 10,000$. All qualitative comparisons are conducted using a fixed representative sample from the FFHQ test set to ensure consistency.

Figure 4 presents the qualitative results of varying the regularization strength, learning rate, and the number of particles. As shown in Figure 4a, a small regularization value ($\tau = 0.5$) overly constrains the latent code to the source distribution, causing the model to retain original features such as masculine traits. The baseline value of $\tau = 5$ achieves a successful attribute transition while effectively preserving the identity of the source image. However, when $\tau$ is excessively large, the sampler fails to maintain the underlying facial structure, leading to unrealistic results.

The learning rate $\eta$ and the number of particles $m$ also play critical roles in the sampling process. As observed in Figure 4b, a smaller learning rate can mitigate artifacts and distortions in the generated images, providing more stable updates. Regarding the number of particles, Figure 4c demonstrates that increasing $m$ consistently improves the quality and stability of the generated samples by providing a better approximation of the gradient flow.

We further analyze the entropy regularization $\epsilon$, which governs the stochasticity and diversity of the generated images. As illustrated in the multi-sample comparison in Figure 4d, low entropy leads to nearly deterministic trajectories with minimal variation between samples. At the intermediate baseline of $\epsilon = 0.1$, the sampler produces diverse yet realistic adversarial candidates. Conversely, increasing $\epsilon$ to 1.0 introduces excessive noise that dominates the drift term, resulting in significant image degradation and loss of semantic content.

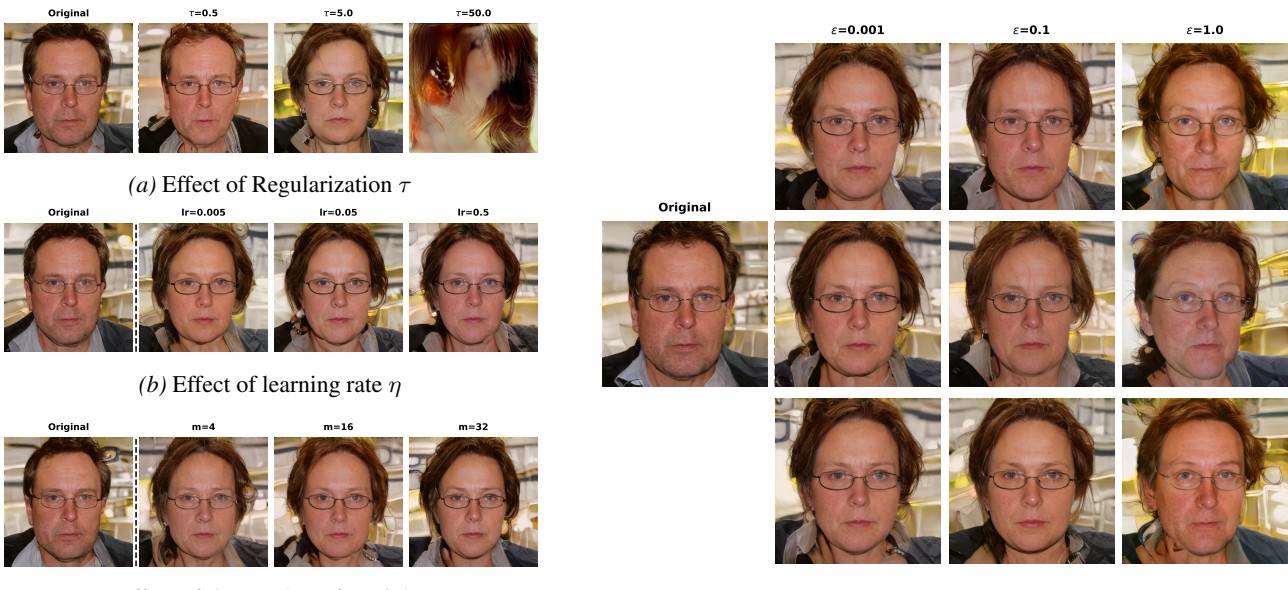

(a) Effect of Regularization $\tau$

(b) Effect of learning rate $\eta$

(c) Effect of the number of particles $m$

(d) Effect of Entropy Regularization $\epsilon$ on Diversity

Figure 4. Qualitative ablation study on the WFR sampler parameters. (a-c) show the sensitivity of single-sample generation to $\tau$, $\eta$, and $m$. (d) visualizes the Top-3 candidates for each $\epsilon$ to demonstrate sample diversity. The leftmost column in each figure represents the original source image.

Finally, we visualize the temporal evolution of the latent code in Figure 5. The progression from iteration 250 to 1,000 demonstrates a stable and smooth transition toward the target attribute, confirming the ability of generating samples from the worst-case distribution of the proposed method.

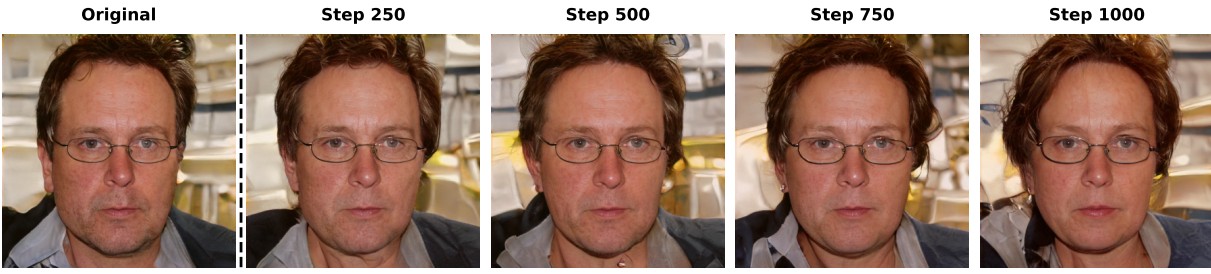

Figure 5. Visualization of the convergence process. The sampler demonstrates a gradual and stable transition towards the target attribute.

## D.2. Multi-class Logistic Regression on CIFAR-10 Features

We assess adversarial robustness using multi-class logistic regression on 512-dimensional features extracted from the CIFAR-10 dataset (Krizhevsky et al., 2009) via a pre-trained ResNet-50. This convex setting is complementary to the deep non-convex experiments in Sections 6.1 and 6.2: it allows us to compare the full set of methods (including RGO-DRO and SVG-DRO) head to head across a sweep of $\epsilon$ values. The model minimizes the multi-class logistic loss

$$\ell(B, x, y) = -y^\top B^\top x + \log\big(\mathbf{1}^\top \exp(B^\top x)\big),$$

where $B \in \mathbb{R}^{512 \times 10}$ is the parameter matrix and $y \in \mathbb{R}^{10}$ is the one-hot label. Perturbations affect only the features $x$; labels are fixed. The transport cost is defined as

$$c((x, y), (x', y')) = \|x - x'\|_2^2 + \infty \cdot \mathbf{1}_{y \neq y'}.$$

To evaluate robustness, we apply a Projected Gradient Descent (PGD) attack (Madry et al., 2018) with $L_2$-norm constraints to the test data. The perturbation magnitude $\Delta$ is normalized by the average $L_2$-norm of the test features, and we vary $\Delta$ from 0 to 0.08. Performance is measured by the misclassification rate.

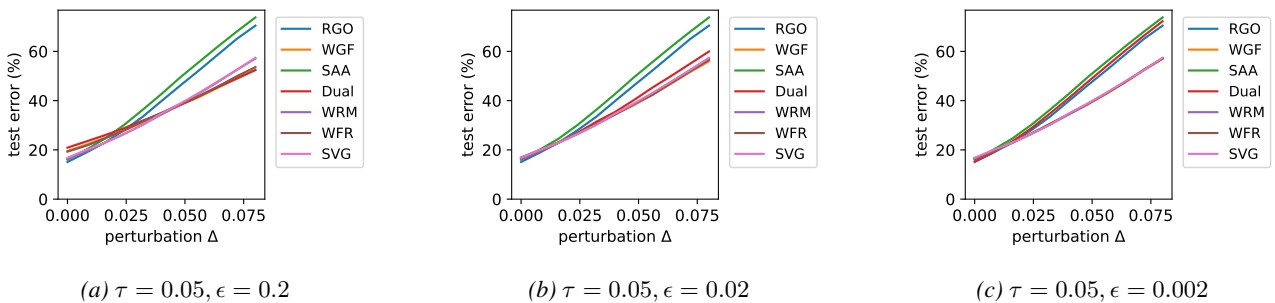

*(a)* $\tau = 0.05, \epsilon = 0.2$        *(b)* $\tau = 0.05, \epsilon = 0.02$        *(c)* $\tau = 0.05, \epsilon = 0.002$

*Figure 6.* CIFAR-10 feature classification error under $L_2$-norm PGD attack. The plots show test error (%) versus normalized perturbation $\Delta$. Models are trained for 10 epochs; inner loops use a stepsize of 0.01 for 100 iterations.

As shown in Figure 6, RGO-DRO fails to provide robustness, performing similarly to the SAA baseline. This suggests the inefficiency of rejection sampling on non-smooth loss functions, which is further supported by the results in Appendix D.4. The Dual method is sensitive to $\epsilon$ and becomes less robust as $\epsilon$ decreases. Notably, SVG-DRO performs nearly identically to WRM, showing a mode collapse issue (detailed in Appendix D.4). As observed in Figure 11, with a small number of particles $m$, the repulsive force fails to prevent convergence to a single adversarial point. In contrast, the SDE-based methods (WFR-DRO and WGF-DRO) avoid this issue and achieve high robustness across all settings.

### D.3. Additional Diagnostics for CNN Experiments

This section provides training details of the experiments in Section 6.2, including convergence curves and wall-clock cost, and a sensitivity analysis with respect to $\tau$ based on the experiments in Section 6.2.

**Experimental setup.** To accommodate the $32 \times 32$ input resolution, we apply the standard CIFAR-10 adaptation of ResNet-18: the first convolution is replaced with a $3 \times 3$, stride-1 kernel, the initial max-pooling layer is removed, and the final fully connected layer is replaced with a 10-way classifier. All methods share this architecture and pretraining. Training uses SGD with learning rate 0.01, momentum 0.9, and weight decay $5 \times 10^{-4}$, batch size 256, for 20 epochs.

**Convergence diagnostics** Figure 7 shows the outer-loop training loss. WRM behaves almost identically to SAA. WGF-DRO, WFR-DRO and Dual exhibit a similar overall trend, but WGF-DRO and WFR-DRO decay faster than Dual, as the Dual gradient estimator suffers exponential bias in $B\tau/\epsilon$ (where $B := \sup \ell(\theta, x)$, see Wang et al. (2025)) when the loss is large. WFR-DRO is consistently a bit faster than WGF-DRO, which is consistent with our theory.

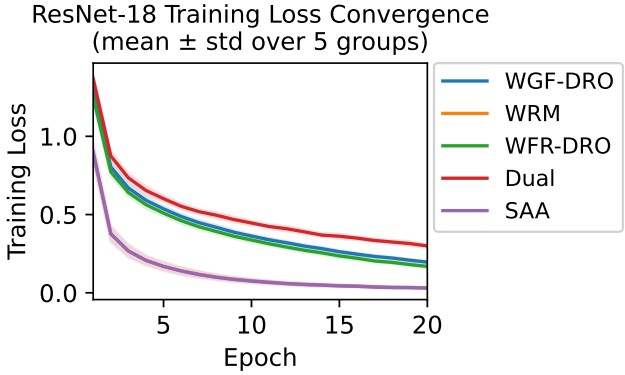

*Figure 7.* Training loss convergence on CIFAR-10 ResNet-18 (mean $\pm$ std over 5 runs).

**Computational cost and $\tau$-sensitivity.** Table 5 reports wall-clock time per epoch; Table 6 reports a $\tau$-sweep on MNIST LeNet-5 under PGD-$L_\infty$ at $\Delta = 0.10$. The overhead relative to WRM scales with $m$; the $m$ particle updates are independent and fully parallelizable on GPU, so the wall-clock overhead is sub-linear in $m = 8$ in practice. The Dual method is cheapest

per epoch among all DRO methods. The $\tau$-sweep confirms WFR-DRO's advantage holds across the full hyperparameter range: WFR-DRO at its worst-case $\tau = 0.05$ (10.92%) still beats WRM at its best $\tau = 1.0$ (12.94%).

*Table 5.* Wall-clock time per epoch (seconds).

| Method | MNIST LeNet-5 | CIFAR-10 ResNet-18 |
|---|---|---|
| SAA | $20.43 \pm 0.56$ | $19.01 \pm 0.22$ |
| Dual | $21.99 \pm 0.89$ | $84.66 \pm 5.63$ |
| WRM | $24.64 \pm 0.69$ | $195.11 \pm 0.65$ |
| WGF-DRO | $32.66 \pm 0.51$ | $711.29 \pm 7.53$ |
| WFR-DRO | $52.57 \pm 1.42$ | $1022.15 \pm 11.69$ |

*Table 6.* $\tau$-sweep error (%) on MNIST LeNet-5 under PGD-$L_\infty$ at $\Delta = 0.10$.

| $\tau$ | WFR-DRO | WGF-DRO | WRM | Dual |
|---|---|---|---|---|
| 0.05 | 10.92 | 12.05 | 13.51 | 11.09 |
| 0.5 | 8.58 | 9.40 | 13.34 | 11.14 |
| 1.0 | 8.78 | 8.83 | 12.94 | 11.36 |

### D.4. Two Moon Classification

In this section, we compare methods on an imbalanced binary classification problem using the 'two moons' dataset. The model is a three-layer neural network with ReLU activations and a hidden dimension of 16. The objective is to minimize the cross-entropy loss. The cost function for samples $(x, y)$ and $(x', y')$ is defined as $c((x, y), (x', y')) = \|x - x'\|_2^2 + \infty \cdot \mathbf{1}_{y \neq y'}$.

The training data is generated by `sklearn.make_moons` with a noise level of 0.1. To introduce class imbalance, the training set of $N_{\text{train}} = 200$ samples consists of 90% positive ($y = 1$) and 10% negative ($y = 0$) samples.

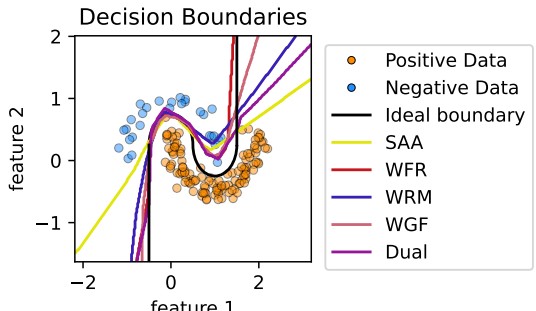

*Figure 8.* Decision boundary comparison for all methods on the two-moon classification task. For all DRO methods, we set $\tau = 0.1$, and for SDRO methods, we set $\epsilon = 0.01$. In each inner loop, WFR and WGF generate $m = 5$ particles.

Figure 8 illustrates the decision boundaries learned by various algorithms against the ideal boundary. Notably, the boundary from the WRM method fails to correctly separate the training data. The Dual method, while separating the classes in high-density regions, learns a boundary that deviates significantly from the ground truth. In contrast, WFR, and WGF learn boundaries that more closely approximate the ideal curve. Among them, WFR and WGF achieve the best results, tracking the ideal separator with high fidelity, which indicates superior performance in this setting.

To evaluate the capacity of generating the worst-case distribution of WRM, WGF, WFR, SVG, and RGO, we conducted an experiment on a pre-trained SAA model. This setup isolates the inner-loop optimization process used to find the worst-case distribution. The evolution of the inner objective function, $\mathbb{E}[-\widetilde{V}_{x,\tau}(z)]$, is depicted in Figure 9.

The efficacy of WGF, WFR, and SVG in approximating the worst-case distribution is demonstrated by a significant increase in the objective value. The reweighting mechanism in WFR contributes to its faster convergence compared to WGF. In contrast, SVG's performance is highly dependent on its initialization parameters. An initial standard deviation of 0.1 leads to slower convergence than a standard deviation of 0.2, which achieves a rate similar to WFR. This sensitivity is attributed to the ability of a larger initial standard deviation to propel particles across the decision boundary, thereby facilitating a more thorough exploration of the perturbation space, as illustrated later in Figure 10. In contrast, WGF and WFR initiate their optimization from the empirical data distribution, which obviates the need to select initialization parameters.

Conversely, WRM and RGO fail to significantly increase the objective. The non-convex nature of the objective, evidenced by the initial dip and subsequent rise for WGF and WFR, likely presents a challenge for the optimization procedures in

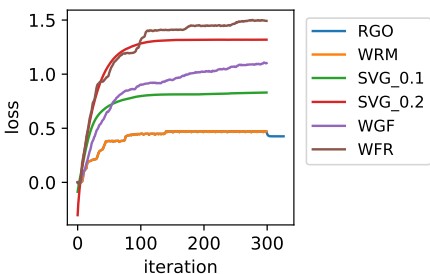

*Figure 9.* Evolution of $\mathbb{E}[\widetilde{V}_{x,\tau}(z)]$. We run all methods for 300 steps with a stepsize of 0.01. For RGO (blue), we run a rejection sampling procedure after solving the inner optimization problem. SVG_0.1 and SVG_0.2 denote initial distributions with a standard deviation of 0.1 and 0.2, respectively.

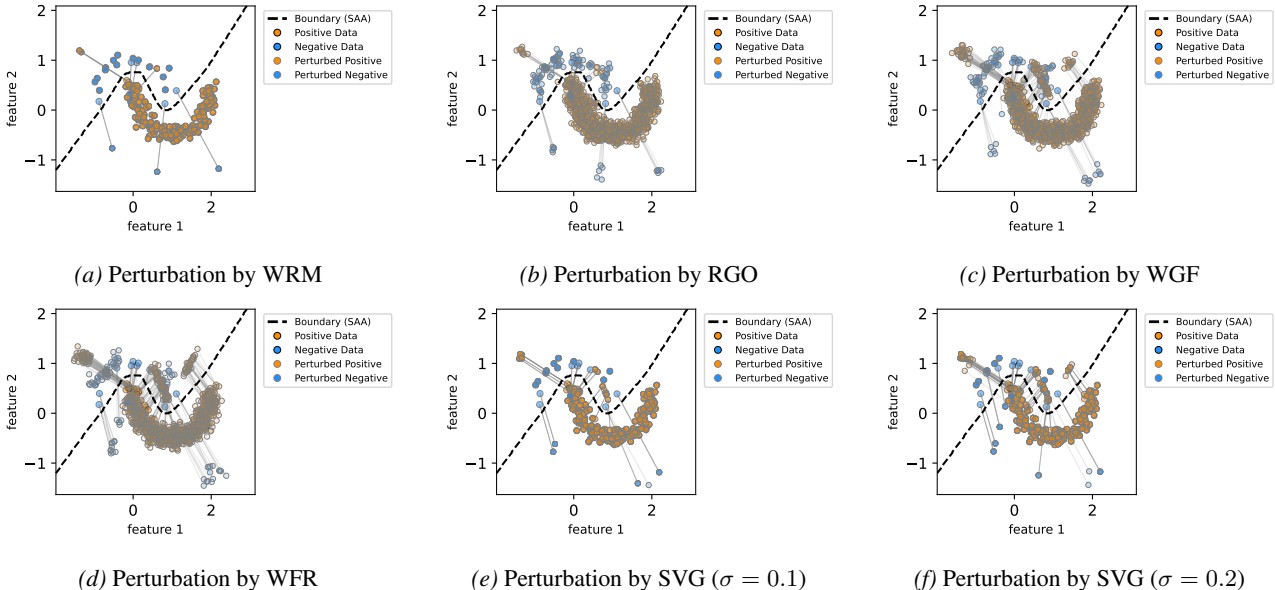

*(a)* Perturbation by WRM  *(b)* Perturbation by RGO  *(c)* Perturbation by WGF

*(d)* Perturbation by WFR  *(e)* Perturbation by SVG ($\sigma = 0.1$)  *(f)* Perturbation by SVG ($\sigma = 0.2$)

*Figure 10.* Visualization of perturbed samples (gray-edged smaller circles) generated from original data (black-edged bigger circles) against the SAA boundary at the final step. For all methods, we use a stepsize of 0.01 and run for 300 iterations. For WFR, the intensity of points visualizes the sample weights. We show perturbations by SVG-DRO with different initializations.

WRM and RGO. Additionally, the performance of RGO is hindered by its rejection sampling stage, which is inefficient for non-smooth objectives.

Figure 10 provides a visual confirmation of these results, displaying the perturbed samples at the final iteration of the inner loop. WRM generates minimal perturbations, with particles remaining in close proximity to the original data points. The perturbations from RGO are qualitatively similar, appearing as slightly noisier versions of the WRM results. In contrast, WGF, WFR, and SVG generate a diverse set of adversarial examples, effectively pushing samples across the decision boundary, including those initially distant from it. Notably, the extent of perturbation for SVG is dependent on the initialization; an initial standard deviation of 0.2 results in more significant perturbations than a standard deviation of 0.1. And the particles generated by SVG for a single data point tend to be highly concentrated.

This phenomenon arises from the interplay between the two forces governing SVGD: a driving force that pushes particles toward regions of higher loss and a repulsive force from the kernel that prevents particle collapse. The optimization dynamics, visualized in Figure 11, show that the driving force initially dominates, causing the particles to converge. As the particles draw closer, the repulsive force increases to counteract this convergence. However, since the number of particles is relatively small, the repulsive force only dominates when particles are very close, leading to the observed particle concentration. This result also explains the reason why SVG shows nearly identical performance as WRM in Appendix D.2.

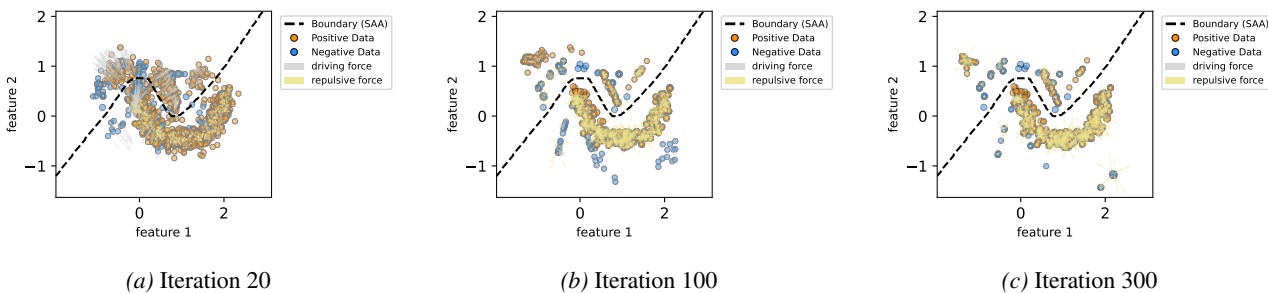

*(a)* Iteration 20       *(b)* Iteration 100       *(c)* Iteration 300

*Figure 11.* Evolution of particle positions in the SVG method. The driving force guides initial convergence, while the repulsive force prevents complete collapse. But in this experiment, the repulsive force fails to push particles apart efficiently, leading to the mode collapse.

## D.5. Classification under Data Imbalance

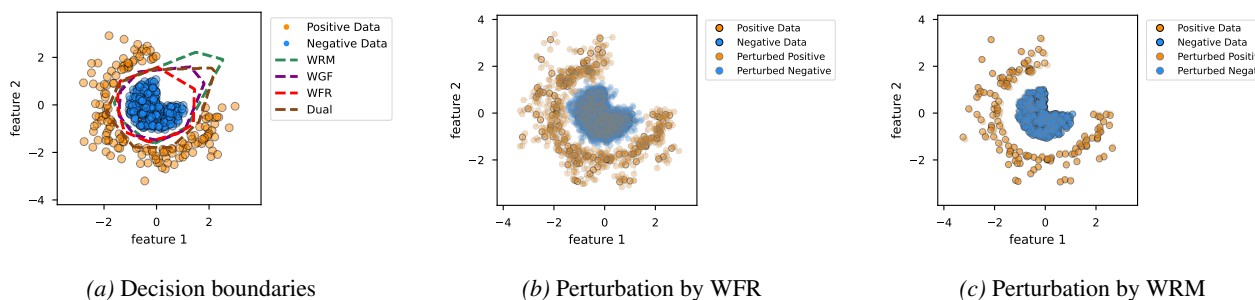

*(a)* Decision boundaries       *(b)* Perturbation by WFR       *(c)* Perturbation by WRM

*Figure 12.* **Robust Decision Boundaries with Biased Data. (a)** Decision boundaries learned by different methods on the biased circle dataset. The training data are shown as orange (positive class) and blue (negative class) points. The final classification boundaries are shown for the WRM (green), WGF (purple), WFR (red), and Dual (brown) models. All models were trained for 40 epochs. We set the regularization parameter $\tau = 2.5$ for all methods and the entropy regularization $\epsilon = 0.15$ for methods based on entropy-regularized Wasserstein DRO problem. **(b)** Samples from the worst-case distribution generated by WFR sampler at the first epoch. **(c)** Worst-case samples generated by WRM method at the first epoch. WRM can only generate discrete distributions as worst-case distribution while entropy-regularized DRO uses potentially continuous distributions as worst-case distribution. The original data points are shown as circles with black edge and the worst-case samples are shown by circles with a shallower color.

This experiment demonstrates the robustness of our proposed methods to data imbalance. The experimental setup is adapted from (Sinha et al., 2018). We generate a dataset where features $X \in \mathbb{R}^2$ are drawn from a Gaussian distribution. The labels are assigned based on the rule $Y = \text{sign}(\|X\|_2 - \sqrt{2})$, which creates two classes separated by a circle of radius $\sqrt{2}$. To establish a clear margin, all data points satisfying $\|X\|_2 \in (\sqrt{2}/1.3, 1.3\sqrt{2})$ are excluded. To simulate a biased training distribution, we remove all samples from the first quadrant. This removal introduces a significant imbalance, testing the ability of each algorithm to learn a generalizable decision boundary rather than overfitting to the biased training data. The model is a neural network with a single hidden layer of 4 units.

Figure 12a visualizes the learned decision boundaries. WFR learns a decision boundary that closely matches the true circular boundary, correctly classifying the held-out data in the first quadrant and thus demonstrating robustness to the distributional shift. In contrast, the WRM boundary is overly expansive, misclassifying large regions of the feature space. This indicates a failure to generalize from the biased training set, resulting in a less reliable classifier. The boundary from WGF lies between those of WFR and WRM, highlighting the benefit of the weight flow mechanism in WFR for achieving a more robust solution. The dual method (Wang et al., 2025) does not perform as well as WFR in this setting. We attribute this to the high sensitivity of the dual method to the choice of hyperparameters ($\epsilon$ and $\lambda$), which can significantly impact the bias of the gradient estimator.

To further investigate the differing behaviors, Figure 12b and Figure 12c visualize the samples generated by WFR and WRM at the first epoch. We observe that the samples generated by WFR algorithm begin to recover parts of the missing data distribution in the first quadrant. Conversely, the samples generated by WRM fail to do so, providing insights into why it learns a less robust boundary.

We also conducted an experiment with an increased training set of 2000 samples while keeping all other settings unchanged. With a larger dataset, the decision boundaries of all methods are expected to converge toward the true circular boundary. In this high-data regime, WGF, Dual, and WRM learn similar boundaries that are nearly circular but still exhibit noticeable deviations. WFR, however, learns a tight and circular boundary, demonstrating its consistent superior performance (see Figure 13).

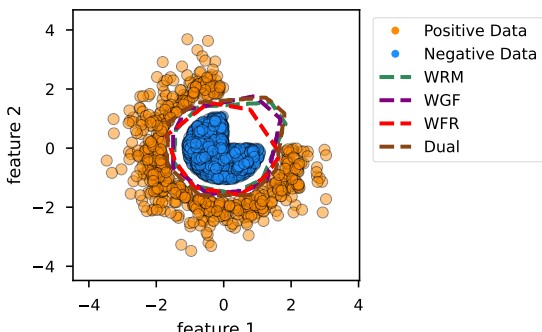

*Figure 13.* Decision boundaries on the Biased Circle dataset with 2000 training samples. WFR still learns a more accurate circular boundary compared to other methods.

### D.6. Uncertain Least Square

We evaluate our methods on a distributionally robust least squares problem, following the setup from (Zhu et al., 2021), which was adapted from (El Ghaoui & Lebret, 1997). The objective is to find a parameter vector $\theta \in \mathbb{R}^{10}$ that minimizes the loss $f_\theta(\xi) = \|A(\xi)\theta - b\|_2^2$, where the system matrix $A(\xi)$ is subject to uncertainty. The matrix $A(\xi) = A_0 + \xi A_1 \in \mathbb{R}^{10 \times 10}$ is an affine function of an uncertain scalar parameter $\xi \in [-1, 1]$. The matrices $A_0, A_1$ and the vector $b$ are fixed, with entries drawn independently from a standard normal distribution $\mathcal{N}(0, 1)$.

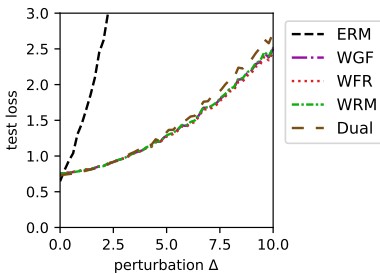

*Figure 14.* Test loss as a function of the perturbation level $\Delta$ for the uncertain least squares problem. All methods were trained for 10 epochs with $\lambda = 0.1$. For methods with an inner loop, the stepsize was 0.0001 for 3000 iterations. SDRO-based methods used $\epsilon = 0.1$. WGF and WFR used $m = 8$ particles.

The training set comprises $N = 10$ samples $\{\xi_i\}_{i=1}^N$ drawn uniformly from $[-0.5, 0.5]$. To evaluate robustness, test samples are drawn from a shifted distribution, specifically uniform on $[-0.5(1 + \Delta), 0.5(1 + \Delta)]$. We vary the shift magnitude $\Delta$ from 0 to 10, where a larger $\Delta$ signifies a greater departure from the training distribution.

Figure 14 shows the test loss as a function of the perturbation level $\Delta$. All distributionally robust methods maintain a significantly lower test loss than the empirical risk minimization baseline, demonstrating their robustness to distributional shifts. At low perturbation levels, all robust methods perform comparably. As $\Delta$ increases, the performance of the dual method degrades more rapidly than the others. WFR shows a marginal improvement over WGF and WRM. This limited advantage is attributable to the one-dimensional, bounded nature of the inner problem, which constrains worst-case samples to lie near the boundary of $[-1, 1]$, thus minimizing the differences between the generated adversarial distributions.

