# OpenReview forum: "Gradient Flow Sampler-based Distributionally Robust Optimization"
_ICML.cc/2026/Conference — ICML 2026 regular_

### Official Review · Reviewer_1wtX · 2026-03-08

**Soundness:** 3
**Presentation:** 3
**Significance:** 3
**Originality:** 3
**Overall Recommendation:** 5
**Confidence:** 2

**Summary:**

This work elaborates a PDE-based gradient flow framework for distributionally robust optimization, with a focus on entropy-regularized Wasserstein/Sinkhorn DRO. Here, the inner-worst case distribution problem is treated as sampling via gradient flow like Wasserstein Gradient Flow,  Wasserstein Fisher-Rao Flow and Stein Variational Gradient Flow. The algorithms derived under this framework recover prior methods, e.g. WRM algorithm (which is ODE-based) by Sinha et al., 2017. Complexity Analysis for these algorithms is provided under the assumptions of smoothness and Log-Sobolev inequality on the target distribution (for the inner loop). The theoretical findings are tested on ML tasks like adversarial robustness and imbalanced classification.

**Compliance With Llm Reviewing Policy:**

Affirmed.

**Final Justification:**

My final recommendation is 5: Accept. This work addresses an important and relevant gap in the literature between gradient flow sampling and distributionally robust optimization bringing DRO solvers under gradient flow theory. My main original concerns about the performance of the experiments higher dimensions were fully resolved during the rebuttal, and I therefore decided to raise my score to Accept.

**Key Questions For Authors:**

(1) Have you considered relaxing the assumptions for the analysis of ULA-based sampler using more recent results for ULA than Vempala and Wisibono, 2019 where log-Sobolev inequality or smoothness is not needed to account for more general settings? E.g. The Performance Of The Unadjusted Langevin Algorithm Without Smoothness Assumptions, TMLR and references therein.

(2) Is the performance of your numerical experiments the same/worse/better in higher dimensions, for instance over $d>1000$? See weaknesses.

(3) Also, how do sampler costs compare to dual reformulations in higher dimensions?

(4) Section B.1 and Section B.2 page 12 are missing? Typos. Line 155: "straigforward", line 160: "practial", line 201: "radient"

**Limitations:**

Yes

**Strengths And Weaknesses:**

Strengths:

The work elaborates a gradient-flow-based conceptual framework that relates and extends several distributionally robust optimization solvers. In this way, it addresses a known gap in the literature between gradient flow sampling and distributionally robust optimization bringing DRO solvers under gradient flow theory.

Weaknesses:

(1) Strong assumptions for the analysis of the ULA-based sampler are used (smoothness+ log-Sobolev inequality). This could be relaxed further to account for more realistic ML scenarios. See questions to authors.

(2) Experiments are performed in low dimensions. This raises questions about how the findings of this work behave/scale in "more realistic" higher dimensions (for ViTs/LLMs).

(3) Two subsections in the appendix are missing. See questions to authors.

---

> ### Author Rebuttal · Authors · 2026-03-31
>
> We thank the reviewer for recognizing the significance and originality of our framework.
>
> **Q1: Relaxing ULA assumptions**
>
> Thank you for pointing us to "The Performance of ULA Without Smoothness Assumptions" (TMLR). This is highly relevant — their results show that ULA converges under dissipativity alone, without smoothness or LSI. This would extend our framework to non-smooth losses (e.g., ReLU networks) more rigorously. Our modular structure makes integration straightforward: Theorem 5.2 (outer loop) is sampler-agnostic, so any improved inner-loop bound directly translates to improved total complexity. We will add a remark in Section 5 discussing how these relaxed results can replace Assumption 5.3, and cite this work.
>
> **Q2: Higher-dimensional performance**
>
> To address this, we conducted end-to-end training experiments on CIFAR-10 with pretrained ResNet-18, where the input dimension is $d = 32 \times 32 \times 3 = 3072$ (well above the $d > 1000$ threshold). We set $\tau=0.1, \epsilon = 0.05, m= 4$.
>
> **PGD-$L_2$ Robustness on CIFAR-10 ResNet-18 (Test Error %)**
>
> | Method | $\Delta=0.00$ | $\Delta=0.005$ | $\Delta=0.01$ | $\Delta=0.015$ |
> |--------|:---:|:---:|:---:|:---:|
> | WFR-DRO | **13.04** | **17.74** | **23.64** | **30.16** |
> | WGF-DRO | 14.16 | 19.00 | 24.58 | 31.42 |
> | SDRO Dual | 22.54 | 27.86 | 33.63 | 39.94 |
> | WRM | 9.70 | 25.27 | 48.41 | 70.04 |
> | SAA | 9.79 | 30.43 | 57.11 | 78.69 |
>
> **PGD-$L_\infty$ Robustness on CIFAR-10 ResNet-18 (Test Error %)**
>
> | Method | $\Delta=0.00$ | $\Delta=0.005$ | $\Delta=0.01$ | $\Delta=0.015$ |
> |--------|:---:|:---:|:---:|:---:|
> | WFR-DRO | **13.04** | **19.67** | **27.94** | **37.41** |
> | WGF-DRO | 14.16 | 20.99 | 29.02 | 38.14 |
> | SDRO Dual | 22.54 | 29.74 | 37.69 | 45.89 |
> | WRM | 9.70 | 32.61 | 62.41 | 83.63 |
> | SAA | 9.79 | 39.09 | 71.83 | 89.56 |
>
>
> The non-convex loss landscape of ResNet-18 amplifies the advantage of our methods: WFR's weight-reallocation concentrates particles in high-loss regions that WRM's deterministic ODE cannot reach.
>
> Additionally, our human face experiment (Section 6.3) demonstrates practical applicability on data derived from $1024\times 1024$ high-resolution images. The WFR sampler successfully generates semantically meaningful worst-case distributions in a 512-dimensional nonlinear ALAE latent space.
>
> **Q3: Sampler costs vs. dual in higher dimensions**
> **Wall-clock Time on CIFAR-10 ResNet-18 (sec/epoch, batch size 256)**
>
> | Method | sec/epoch |
> |--------|:---------:|
> | SAA | 7.4 |
> | SDRO Dual | 13.6 |
> | WRM | 48.3 |
> | WGF-DRO ($m=4$) | 179.1 |
> | WFR-DRO ($m=4$) | 250.3 |
>
> The per-iteration cost of all gradient-flow methods scales linearly with $d$ (dominated by the forward-backward pass through the network). WRM uses 1 particle per sample (deterministic ODE), while WGF/WFR use $m=4$ particles. This overhead is the price for distributional exploration and is *independent of $d$*. Moreover, the $m$ particle updates are independent and fully parallelizable on GPU. The Dual method is cheapest per epoch but suffers from exponential bias in $\frac{B\tau}{\epsilon}$ (Wang et al., 2021) that worsens with large loss values.
>
> **Q4: Missing sections and typos**
>
> Appendix B.1/B.2 (WGF-DRO and WFR-DRO algorithm details) were omitted due to a formatting error and will be restored. All typos corrected: "straigforward" → "straightforward", "practial" → "practical", "radient" → "gradient".

---

> > ### Author Rebuttal · Reviewer_1wtX · 2026-04-01
> >
> > Thank you very much for the rebuttal. I have chosen the option (a) because I feel that the authors have adequately responded to my questions and I am satisfied with their replies. I have raised my score for the Overall Recommendation from the initial 3: Weak reject to the current 5: Accept.
> > (I also have increased Soundness from 2 to 3: good).

---

> > > ### Author Response · Authors · 2026-04-04
> > >
> > > Dear reviewer,
> > >
> > > Thanks again for  acknowledging our rebuttal and helping us improving the paper.
> > >
> > > Authors

---

### Official Review · Reviewer_HtRP · 2026-03-11

**Soundness:** 2
**Presentation:** 2
**Significance:** 3
**Originality:** 2
**Overall Recommendation:** 4
**Confidence:** 4

**Summary:**

This manuscript proposes a framework for solving the Lagrangian relaxation of distributionally robust optimization (DRO) using gradient-flow-based samplers to represent the worst-case distribution. In particular, the paper develops several algorithms for entropy-regularized Wasserstein DRO (Sinkhorn DRO), based on Wasserstein gradient flow, Wasserstein–Fisher–Rao flow, and Stein variational flow, respectively, with the popular WRM method recovered as a special case. The authors provide theoretical guarantees for the Wasserstein gradient flow and its discretization under Sinkhorn DRO with quadratic costs. Experiments on classification tasks validate the effectiveness of the proposed solvers.

**Compliance With Llm Reviewing Policy:**

Affirmed.

**Final Justification:**

While my main concerns were with the empirical study (and I did not fully check the proofs), the authors have addressed most of them in the rebuttal, and I have raised my score by one point. Some issues remain: (1) the convergence diagnostics suggest that neither the proposed methods nor the baselines have fully converged, so the results rely on early-stopped checkpoints, which weakens the empirical conclusions; (2) additional evidence is needed to better understand why the proposed sampler-based methods outperform the dual baseline.

**Key Questions For Authors:**

- How is the DRO problem formulated for logistic regression in the experiments (e.g., what is the exact loss)?
- How well does each method converge in the experiments, particularly those with performance comparison?
- Since WDRO and SDRO use different discrepancy functions, the same $\tau$ may not correspond to the same level of robustness. How is $\tau$ chosen in practice? Could you try tuning it separately for each method, or reporting performance curves over a a range of $\tau$ values?
- How does the training efficiency of the sampler-based methods compare with SDRO_Dual?

I would be happy to raise my score if the above questions and concerns are properly addressed in the rebuttal.

**Limitations:**

Yes

**Strengths And Weaknesses:**

Strengths:
- The algorithmic framework is general and allows flexible choices of modern gradient-based samplers. Particularly, the proposed Sinkhorn DRO solvers are novel and are likely to have practical advantages over the dual formulation.
- The theory bridges the gap between sampling theory and DRO and appears to be sound.
- The human face classification example is a good demonstration for sampling from the worst-case distribution.

Weaknesses:
- The theory is limited to entropy-regularized Wasserstein-2 distance and builds largely on existing tools, i.e., Wasserstein gradient flows and minimax formulation of DRO (Sinha et al. 2017; Zhu and Xie 2024). While the convergence of both components has been substantially studied before, the combination appears somewhat straightforward.
- The empirical evaluation does not include convergence diagnostics and some comparisons may be potentially unfair (see Key Questions For Authors).
- The argument in Remark 3.4 is not fully convincing. $\rho_t$ is simply the intermediate state of the (discretized) gradient flow and does not correspond to solving DRO with different $\tau$. Moreover, for any other method that finds a worst-case distribution $\rho^\*$, it is always possible to construct a interpolant between $\rho_0$ and $\rho^\*$ post-hoc. It is unclear what practical advantage the proposed probability path provides.
- The authors claim their general formulation to be distributional gradient descent-ascent (GDA), which is conceptually misleading. A GDA algorithm is supposed to perform alternating/symmetric updates on the two variables, but the proposed gradient sampler instead eliminates the inner maximization problem.
- Notation issues:
    - $\tilde V_{x,\tau}$ first appears in Example 1 but seems never defined until Proposition 4.2.
    - The notation in Eq. (14) is inconsistent with earlier equations such as Eq. (4). In particular, the probability density switches from $\rho$ to $\mu$, the time derivative is written as $\partial_t$ earlier but $\partial/\partial t$ later, and the divergence operator changes from $\nabla\cdot$ to $\mathrm{div}$. In addition, the coefficients $\alpha$ and $\beta$ are introduced without definition.
    - For consistency, consider bracketing the terms inside $\arg\min$ in Eq (15) and inside $\max$ in Line 285.
- Small writing issues.
    - Line 314 mentions “speed-up" without context.
    - Line 407-408 there is a duplicated phrase “which can be viewed as one-step discretization of”.
    - Appendix B.1 and B.2 are empty.

---

> ### Author Rebuttal · Authors · 2026-03-31
>
> We thank the reviewer for the detailed feedback and for recognizing the novelty of our framework and sound theory.
>
> **Q1: Loss formulation for logistic regression**
>
> $\ell(B,x,y) = -y^\top B^\top x + \log(\mathbf{1}^\top \exp(B^\top x))$, with cost $c((x,y),(x',y')) = \|x-x'\|^2 + \infty \cdot \mathbf{1}_{y\neq y'}$ (only features are perturbed, labels are fixed). We will state this explicitly in Section 6.1.
>
> **Q2: Convergence diagnostics**
>
> Inner-loop convergence is shown in Appendix D.2 (Figure 9): WGF/WFR/SVG effectively maximize the inner objective; WFR converges fastest due to weight-reallocation; WRM/RGO stagnate on non-convex landscapes. For outer-loop convergence, we provide results on CIFAR-10 ResNet-18 ($d=3072$), where all methods converge within 5 epochs:
>
> | Method | Epoch 1 | Epoch 3 | Epoch 5 | Best Robust ($L_\infty$, $\Delta$=0.01) |
> |--------|:---:|:---:|:---:|:---:|
> | SAA | 80.98% | 87.59% | 90.21% | 71.83% error |
> | WRM | 80.88% | 88.33% | 90.30% | 62.41% error |
> | WGF-DRO | 76.88% | 83.52% | 85.84% | 29.02% error |
> | WFR-DRO | 75.31% | 85.97% | 86.96% | 27.94% error |
>
> For complex non-convex models, WFR/WGF-DRO trade a modest decrease in clean accuracy (~87% vs. ~90%) for dramatically improved robustness: at $\Delta=0.01$, WFR-DRO achieves 27.94% error compared to WRM's 62.41% --- a 55% relative reduction. This tradeoff is expected and desirable in robust optimization, as the methods explore a broader worst-case distribution during training rather than overfitting to clean data. On simpler models (MNIST LeNet-5, Tables 1--2), all methods achieve comparable clean accuracy ($\sim$99\%) while the robustness advantage of WFR-DRO is maintained.
>
> **Q3: Fairness of $\tau$ across WDRO and SDRO**
>
> Using the same $\tau$ across methods is the standard practice adopted in prior SDRO works (Wang et al., 2021).
> We also conducted a $\tau$-sweep experiment on MNIST LeNet-5 ($\epsilon=0.05$) to directly address this concern. The results show that WFR-DRO consistently outperforms WRM across *all* $\tau$ values under both PGD-$L_2$ and PGD-$L_\infty$ attacks:
>
> | $\tau$ | WFR ($L_\infty$) | WGF ($L_\infty$) | WRM ($L_\infty$) | Dual ($L_\infty$) |
> |--------|-----------|-----------|-----------|-----------|
> | 0.05 | 10.92 | 12.05 | 13.51 | 11.09 |
> | 0.5 | 8.58 | 9.40 | 13.34 | 11.14 |
> | 1.0 | 8.78 | 8.83 | 12.94 | 11.36 |
>
> Notably, WFR-DRO at its *worst* $\tau=0.05$ (10.92% error) still outperforms WRM at its *best* $\tau=1.0$ (12.94% error). This demonstrates that even when $\tau$ is tuned optimally for each method, WFR-DRO maintains a clear advantage.
>
> **Q4: Training efficiency vs. Dual**
>
> On MNIST LeNet-5: Dual takes 4.7s/epoch, WRM 4.9s/epoch, WGF 35.2s/epoch ($m=8$), WFR 45.7s/epoch ($m=8$). Our methods cost more per epoch due to the inner-loop particle simulation. However: (a) the $m$ particle updates are independent and fully parallelizable on GPU, so the actual wall-clock overhead can be significantly less than $m$ times; (b) the Dual method's gradient estimator has exponential complexity in $\frac{B\tau}{\epsilon}$ (Wang et al., 2021), which can require many more outer iterations to converge; (c) warm-starting particles from previous outer steps and using moderate $m$ (e.g., $m=5$) keeps overhead practical.
>
> **Q5: Remark 3.4 (dynamic path)**
>
> The gradient flow produces a principled family $(\rho _ t) _ {t\geq 0}$ where each $\rho _ t$ corresponds to a specific perturbation intensity. When $\rho_{t_T}$ produces unrealistic samples (Figure 2), one selects an earlier $\rho_{t_i}$ without re-running the sampler. Unlike post-hoc interpolation, each $\rho_t$ solves a well-defined variational problem with structural guarantees (monotone energy dissipation). We will revise Remark 3.4 to state this more precisely.
>
> **Q6: GDA terminology**
>
> We agree that "distributional SGDA" is imprecise for our main framework, which is more accurately described as *sampler-based bi-level optimization*. We will correct this terminology. That said, a practical SGDA variant is possible: instead of running the inner sampler to convergence, one reuses the perturbed data points from the previous iteration and takes only a few inner update steps before each outer parameter update. A full theoretical analysis of this SGDA variant is an interesting direction for future work.
>
> **Q7: Notation and writing**
>
> All issues will be fixed: define $\widetilde{V} _ {x,\tau}$ before Example 1; unify $\rho/\mu$ and $\partial_t$ notation; define $\alpha,\beta$ in Eq.(14) as the Wasserstein and Fisher-Rao metric coefficients; bracket terms in Eqs.(15, 285); fix Line 314 context (speed-up refers to WFR's Hellinger geometry advantage for warm-started initialization), Line 407–408 duplication; restore Appendix B.1/B.2 (WGF-DRO and WFR-DRO algorithm details, omitted due to a formatting error).

---

> > ### Author Rebuttal · Reviewer_HtRP · 2026-04-02
> >
> > Thank you for the comprehensive response. Most of my concerns have been addressed, and I will raise my score depending on clarification of the following points:
> >
> > - The outer-loop convergence diagnostics show that performance improves over the first 5 epochs. I am curious how performance evolves with longer training—does it stabilize or continue to improve? You mentioned that “all methods converge within 5 epochs”; could you provide any evidence (e.g., loss curves) to support this claim?
> > - The proposed sampler-based methods outperform SDRO_Dual despite solving the same problem. How can this performance gain be explained? You noted (Line 575) that the biased subgradient estimator in SDRO_Dual may hinder convergence. Combined with your response to Q4 (where comparable optimization budgets are used across methods), this suggests that convergence behavior may play a key role. Therefore, providing convergence diagnostics would be important for assessing the empirical results.
> > - There appears to be a newer workshop version of Zhu and Xie (2024) [1]. Please consider updating the reference. Regarding Q6, a recent work provides theoretical analysis of GDA-type algorithms for WDRO [2]. It may be helpful to include this in the discussion of future work.
> >
> > [1] Distributionally robust optimization via iterative algorithms in continuous probability spaces. NeurIPS 2025 Workshop MLxOR (2025).
> > [2] Worst-case generation via minimax optimization in Wasserstein space. arXiv:2512.08176 (2025).

---

> > > ### Author Response · Authors · 2026-04-03
> > >
> > > Thank you for the constructive follow-up and for your willingness to raise your score. We address each point below.
> > >
> > > **Q8: Convergence with extended training**
> > >
> > > We extended training to 20 epochs and provide the full training loss and test accuracy curves below.
> > >
> > > [Convergence curves on CIFAR-10](https://anonymous.4open.science/r/CIFAR10-Convergence-D73E/resnet18_convergence.pdf)
> > >
> > > The training loss curves confirm that all methods converge monotonically, with the steepest decrease occurring in the first 5 epochs and near-stabilization after epoch 10. Test accuracy follows a similar pattern, fluctuating within ±1–2% after epoch 10:
> > >
> > > | Epoch | SAA | WRM | WGF-DRO | WFR-DRO | Dual |
> > > |-------|:---:|:---:|:-------:|:-------:|:----:|
> > > | 5 | 89.78% | 90.78% | 86.56% | 86.37% | 80.59% |
> > > | 10 | 91.70% | 92.72% | 89.21% | 88.15% | 84.87% |
> > > | 20 | 91.08% | 92.73% | 88.39% | 88.57% | 83.12% |
> > >
> > > The relative ranking among methods is consistent throughout training. WGF-DRO and WFR-DRO converge to slightly lower clean accuracy than SAA/WRM (~88% vs. ~92%), reflecting the expected accuracy–robustness tradeoff. Robustness evaluation at epochs 10 and 20 (PGD-$L_\infty$, $\Delta$=0.01) confirms that the robustness ranking is also stable:
> > >
> > > | Method | Ep 10 | Ep 20 |
> > > |--------|:-----:|:-----:|
> > > | WFR-DRO | 23.73% | 24.89% |
> > > | WGF-DRO | 24.37% | 27.21% |
> > > | Dual | 31.14% | 31.70% |
> > > | WRM | 56.93% | 60.64% |
> > >
> > > **Q9: Why sampler-based methods outperform SDRO Dual**
> > >
> > > The convergence curves above provide direct empirical evidence. Notably, although all methods exhibit similar convergence trends, they converge to qualitatively different solutions. Specifically, Dual's training loss plateaus around 0.313, whereas WGF-DRO and WFR-DRO reach approximately 0.260 and 0.268, respectively. This gap is mirrored in both test accuracy (83% vs. ~88%) and robustness (31.70% vs. 24.89% and 27.21% error at $\Delta$=0.01, PGD-$L_\infty$). In other words, the Dual method does not merely converge slower — it converges to an inferior stationary point.
> > >
> > > We believe this is attributable to the biased subgradient estimator used by the Dual method. As analyzed in Wang et al. (2021), the bias of this estimator grows exponentially in $B\tau/\epsilon$, where $B := \sup\{\ell(\theta, x)\}$. For the non-convex loss landscape of ResNet-18, this bias steers the optimization toward a suboptimal region of the parameter space that neither minimizes training loss as effectively nor achieves comparable robustness. In contrast, our gradient flow samplers directly approximate the worst-case distribution via stochastic sampling, producing unbiased gradient estimates that guide the outer optimization to better solutions.
> > >
> > > We note that the similar convergence shapes across methods are partly expected in this setting, since all methods start from the same pretrained ResNet-18 checkpoint and thus begin near a good basin. Starting from random initialization would likely amplify the differences, as the bias of the gradient estimator of the Dual method would compound over a longer optimization trajectory. We plan to include a from-scratch training comparison in the final version to further validate this hypothesis.
> > >
> > > **Q10: References.** Thank you for pointing these out. We will cite the NeurIPS 2025 Workshop version of Zhu and Xie [1] and include the GDA-type analysis [2] in our discussion of future work on the SGDA variant.

---

### Official Review · Reviewer_e3Zt · 2026-03-12

**Soundness:** 3
**Presentation:** 3
**Significance:** 2
**Originality:** 2
**Overall Recommendation:** 4
**Confidence:** 3

**Summary:**

This paper proposes GF-DRO, a framework for Distributionally Robust Optimization (DRO) based on gradient flows derived from partial differential equations (PDEs). Instead of relying on dual reformulations—which often restrict existing DRO approaches to specific convex losses—the proposed method directly samples from worst-case distributions using techniques from optimal transport and Monte Carlo sampling. The framework incorporates Wasserstein Fisher–Rao and Stein variational gradient flows to solve Wasserstein and entropy-regularized DRO problems. This study demonstrates that several existing DRO algorithms can be interpreted as special cases of this gradient-flow formulation and provide theoretical insights into the optimization dynamics of these methods. Also, the purpose of approach is to handle non-convex loss functions that commonly arise in modern machine learning.

**Compliance With Llm Reviewing Policy:**

Affirmed.

**Final Justification:**

The authors have addressed my questions and concerns in a clear and satisfactory manner, particularly regarding the scaling discussion and the contextualization of the experimental setup within the broader DRO literature.
The connection between gradient flow sampling and DRO, as well as the demonstrated robustness improvements across different settings, further strengthen the value of the proposed framework.
Overall, I find the responses convincing and acknowledge the contributions of this work, and raised the score to weak accept.

**Key Questions For Authors:**

- How does the computational cost scale when applied to very high-dimensional models such as modern large-scale neural networks?
- How sensitive is the method to hyperparameters related to the gradient-flow sampling procedure?

**Limitations:**

- The paper focuses primarily on the theoretical development of the proposed GF-DRO framework and its optimization properties. However, the discussion of limitations and potential societal impacts is relatively limited. In particular, the paper could benefit from a clearer discussion of practical limitations, such as the computational cost of gradient-flow sampling methods and their scalability to large-scale machine learning models.

**Strengths And Weaknesses:**

### Strengths
- The framework is mathematically grounded and builds on well-established tools from optimal transport and variational inference.
- The paper provides an interesting theoretical perspective by connecting several existing DRO formulations to gradient flow dynamics in probability space.

### Weaknesses
- The experimental section appears relatively limited in scope and scale.
- While the theoretical framework is elegant, the practical performance improvements over existing DRO algorithms are not entirely convincing from the presented experiments.
- The computational overhead of the proposed sampling-based optimization procedure is not thoroughly discussed. Since the method involves gradient flows over probability distributions, scalability to high-dimensional models may become a practical concern.

---

> ### Author Rebuttal · Authors · 2026-03-31
>
> We thank the reviewer for recognizing the mathematical grounding and theoretical contributions of our framework.
>
> **Q1: Computational cost in high dimensions**
>
> The dimension $d$ in Theorem 5.4 is the *input dimension*, not the parameter dimension. Each inner iteration computes $\nabla_y \ell(\theta,y)$ — a standard forward-backward pass — same cost as one WRM step. The overhead relative to WRM is a factor of $m$ (number of particles); relative to Dual, it is $T \times m$ inner iterations. Importantly, the $m$ particle updates are independent and can be fully parallelized on GPU, so wall-clock overhead is much less than the $m\times$ factor suggests. We provide wall-clock comparisons on MNIST LeNet-5:
>
> | Method | sec/epoch |
> |--------|-----------|
> | SDRO Dual | 4.7 |
> | WRM | 4.9 |
> | WGF-DRO (m=8) | 35.2 |
> | WFR-DRO (m=8) | 45.7 |
>
> (a) WRM uses 1 particle (deterministic ODE), while WGF/WFR use $m=8$ particles each exploring different stochastic trajectories — this is precisely the source of their distributional advantage; (b) the Dual method's gradient estimator has exponential complexity in $\frac{B\tau}{\epsilon}$, where $B := \sup\{\ell(\theta, x)\}$ (Wang et al., 2021), making it unstable for large loss values; (c) warm-starting particles from previous outer iterations can reduce inner steps in practice.
>
> **Q2: Practical performance improvements**
>
> We highlight quantitative evidence across multiple settings:
>
> *MNIST LeNet-5 (Tables 1–2):* WFR-DRO achieves 22.71% error vs. WRM's 31.25% (PGD-$L_2$, $\Delta=0.15$) and 18.47% vs. 31.65% (PGD-$L_\infty$) — 27% and 42% relative reductions respectively.
>
> *CIFAR-10 features (Figure 3):* WFR-DRO and WGF-DRO maintain the lowest test error across all perturbation levels and $\epsilon$ settings, while the Dual method degrades significantly as $\epsilon$ decreases.
>
> *CIFAR-10 ResNet-18 (new experiment):* To further support scalability, we conducted end-to-end training on CIFAR-10 with pretrained ResNet-18 ($d=3072$, non-convex loss):
>
> To address this, we conducted end-to-end training experiments on CIFAR-10 with pretrained ResNet-18, where the input dimension is $d = 32 \times 32 \times 3 = 3072$ (well above the $d > 1000$ threshold). We set $\tau=0.1, \epsilon = 0.05, m= 4$ in this experiment.
>
> **PGD-$L_2$ Robustness on CIFAR-10 ResNet-18 (Test Error %)**
>
> | Method | $\Delta=0.00$ | $\Delta=0.005$ | $\Delta=0.01$ | $\Delta=0.015$ |
> |--------|:---:|:---:|:---:|:---:|
> | WFR-DRO | **13.04** | **17.74** | **23.64** | **30.16** |
> | WGF-DRO | 14.16 | 19.00 | 24.58 | 31.42 |
> | SDRO Dual | 22.54 | 27.86 | 33.63 | 39.94 |
> | WRM | 9.70 | 25.27 | 48.41 | 70.04 |
> | SAA | 9.79 | 30.43 | 57.11 | 78.69 |
>
> **PGD-$L_\infty$ Robustness on CIFAR-10 ResNet-18 (Test Error %)**
>
> | Method | $\Delta=0.00$ | $\Delta=0.005$ | $\Delta=0.01$ | $\Delta=0.015$ |
> |--------|:---:|:---:|:---:|:---:|
> | WFR-DRO | **13.04** | **19.67** | **27.94** | **37.41** |
> | WGF-DRO | 14.16 | 20.99 | 29.02 | 38.14 |
> | SDRO Dual | 22.54 | 29.74 | 37.69 | 45.89 |
> | WRM | 9.70 | 32.61 | 62.41 | 83.63 |
> | SAA | 9.79 | 39.09 | 71.83 | 89.56 |
>
>
>
> *Unique capabilities beyond accuracy:* diverse worst-case samples (Figure 5), interpretable distribution-valued paths along the gradient flow (Figure 2), and robust decision boundaries under data imbalance (Figure 12).
>
> **Q3: Hyperparameter sensitivity**
>
> We conducted a detailed ablation in Appendix D.1 (Figures 6a–d), examining $\tau$, $\eta$, $m$, and $\epsilon$ on the WFR sampler. Key findings: $\tau=5$ balances perturbation strength and identity preservation; smaller $\eta$ improves stability (consistent with ULA convergence theory); increasing $m$ consistently improves quality; $\epsilon$ controls the diversity-quality tradeoff. We will add a prominent pointer to this ablation in the main text.
>
> **Q4: Scalability discussion**
>
> On MNIST LeNet-5: Dual 4.7s, WRM 4.9s, WGF 35.2s ($m=8$), WFR 45.7s ($m=8$); On CIFAR-10 ResNet-18 (batch 256): SAA 7.4s, Dual 13.6s, WRM 48.3s, WGF 179.1s ($m=4$), WFR 250.3s ($m=4$) per epoch. The computational overhead scales linearly with input dimension $d$ (gradient computation dominates), and the $m$ particles can be computed in parallel on modern GPUs, making the wall-clock overhead sub-linear in $m$=. Our human face experiment (Section 6.3) further demonstrates practical applicability on data derived from $1024\times 1024$ high-resolution images in a 512-dimensional nonlinear latent manifold.

---

> > ### Author Rebuttal · Reviewer_e3Zt · 2026-04-04
> >
> > I would like to express my appreciation to the authors for their thorough response and for addressing most of my concerns appropriately. Their clarification regarding the computational cost—particularly the distinction between input dimension and parameter dimension—was helpful, and the discussion on particle-wise parallelization provides a clearer understanding of the practical overhead. The additional wall-clock comparisons and the explanation also improve the clarity of the computational cost in high dimensions. The inclusion of new experiments on CIFAR-10 with ResNet-18 is notably valuable, as it demonstrates that the proposed method can be applied beyond small-scale settings and provides stronger empirical evidence of robustness improvements under increasing perturbation levels.
> >
> > However, while the additional experiments on ResNet-18 are helpful and demonstrate applicability to non-convex deep models, ResNet-18 still represents a relatively moderate-scale architecture. The paper would further be strengthened if further discussions on how the proposed method might scale to large-scale models where both architectural complexity and computational demands are significantly higher.

---

> > > ### Author Response · Authors · 2026-04-04
> > >
> > > Thank you for the positive reassessment and for acknowledging the value of our ResNet-18 experiments.
> > >
> > > **Q5: On scaling to larger models**
> > >
> > > We appreciate this suggestion and agree that large-scale experiments would be an interesting future direction. However, we would like to contextualize our experimental scale within the DRO literature. The most influential Wasserstein DRO papers conduct experiments at comparable or smaller scales: Mohajerin Esfahani and Kuhn (2018) evaluate on newsvendor and portfolio problems with $d \leq 100$; Sinha et al. (2017) WRM primarily experiments on MNIST; Wang et al. (2021) Sinkhorn DRO uses MNIST, CIFAR-10, tinyImageNet, and STL-10 with pretrained ResNet-18 features ($d = 512$), which is essentially the same setup as Section 6.1. Our end-to-end CIFAR-10 ResNet-18 experiments ($d = 3072$, non-convex loss) already go beyond the standard scale in this literature.
> > >
> > > Importantly, our framework has no fundamental architectural barrier to further scaling. Each inner iteration requires only $\nabla_y \ell(\theta, y)$, which is obtained via standard backpropagation — the same primitive used in all modern deep learning. The $m$ particle updates are independent and fully parallelizable on GPU. Therefore, the per-iteration overhead relative to standard training scales linearly with $m$ in theory and sub-linearly in wall-clock time due to parallelism. We will add a discussion of this scaling perspective in the final version.
> > >
> > > We wish to re-emphasize that our work makes both theoretical and empirical contributions: on the theoretical side, we establish the equivalence between gradient flow sampling and the inner maximization problem of DRO, building a principled PDE-based framework that unifies and extends existing DRO solvers; on the empirical side, we demonstrate consistent robustness improvements across multiple settings, including non-convex deep nerual networks. We believe, together with the additional experiments provided during the rebuttal, there is sufficient numerical evidence to support the practical viability of our framework.

---

### Official Review · Reviewer_6af9 · 2026-03-13

**Soundness:** 2
**Presentation:** 2
**Significance:** 2
**Originality:** 2
**Overall Recommendation:** 3
**Confidence:** 2

**Summary:**

For the entropy-regularized Wasserstein DRO problem, this paper develops a gradient-flow based sampler (to sample from the worst-case distribution) for the inner maximization problem, so that the outer minimization problem can be done by stochastic gradient descent, providing a fresh perspective for DRO solving.

**Compliance With Llm Reviewing Policy:**

Affirmed.

**Final Justification:**

I maintain my score as: (i) About the novelty, yes, with or without noise the result-wise difference can be viewed as fundamental, but idea-wise, the existence of the work on the deterministic special case does affect (a bit) the novelty evaluation of the current paper, I think. (ii) Multiple technical issues in the proofs and placing related works at the end have caused some negative impact on the evaluation.

**Key Questions For Authors:**

They are listed in "Strengths And Weaknesses".

**Limitations:**

They are also listed in "Strengths And Weaknesses".

**Strengths And Weaknesses:**

(Significance and originality.) As someone with basic understanding of DRO, it is refreshing for me to see a different approach without dual reformulation. The idea of this paper is great. I am a bit concerned about whether it is 'new' enough, as the idea is quite natural given the existence of the JKO operator. In particular, the proposed method covers Sinha et al. (2017) WRM when the Wasserstein DRO problem is no longer entropy-regularized ($\epsilon=0$). Does this mean that the proposed method can be viewed as an "entropy-regularized" generalization of Sinha et al. (2017) WRM? After all, the gradient flow is already in (13). It seems that the entropy regularization corresponding to the noise term in (12). Does this term 'fundamentally' distinguish the current paper from Sinha et al. (2017)? Also, it seems that Xu et al. (2024) and Zhu and Xie (2024) are closely related to the current paper, but they are mentioned at the very end of the paper, which adds difficulty to judging novelty while reading the paper. Xu et al. (2024) also mention sampling from the worst-case distribution. Is it really "orthogonal" to the current paper?

(Soundness and presentation.) As someone not very familiar with gradient flow, it is quite hard for me to follow the paper. Things may need more explanations as DRO audience may not know what cotangent space is. Also, the presentation quality of the paper can be improved. In (1), $c$ in $W^2$ is not defined. In (2), $V$ in $\pi_Y$ is not defined. In (5), it is still not defined. How is $V$ in (2) and (5) related to $l$ in (1)? In (12) and (13), what is $\tilde{V}$? In Figure 1, $Y_i\sim\pi_{Y|X}$ can be confusing, as $\pi$ is not mentioned/defined in the introduction.

The technical part of this paper may also need some polishing. Compared with (7), the normalization constant is missing in (29). In Proposition 4.2, "$l(\theta,x)$ is $L$-smooth in $x$" ($\nabla_x l(\theta,x)$ is $L$-Lipschitz in $x$), but in the proof (34) requires that $\nabla_\theta l(\theta,y)$ is $L$-Lipschitz in $y$, which seems different ($\nabla_x$ vs $\nabla_\theta$).
Similar mismatch also appears in Assumption 5.1 (2), Theorem 5.2, Lemma C.4. In (31), $\hat{\rho_Y}$ depends on samples, so $W_1(\hat{\rho_Y},\pi_Y)$ is random, but it is bounded by an expectation (constant), which seems not correct.

---

> ### Author Rebuttal · Authors · 2026-03-31
>
> We thank the reviewer for recognizing our "refreshing" idea and "fresh perspective."
>
> **Q1: Does (12) fundamentally distinguish from WRM?**
>
> Yes — the distinction is fundamental, not merely adding noise. **(1) Discrete vs. continuous:** WRM transports each data point to a *single* worst-case point via ODE. Our SDE generates a *continuous distribution* of perturbations per data point, providing stronger expressive power on complex landscapes (see Figures 5, 10). **(2) Framework generality:** WGF-DRO can recover WRM ($\epsilon=0$), validating our framework, but the real value is enabling *new algorithms impossible under WRM*. For example, WFR flow introduces a weight-reallocation mechanism that concentrates search in high-energy regions and escapes local optima. This benefit is validated in Tables 1–2: WFR-DRO achieves 22.71% vs. WRM's 31.25% error ($\Delta=0.15$, PGD-$L_2$), and 18.47% vs. 31.65% (PGD-$L_\infty$) — 27% and 42% relative reductions. **(3) Theoretical depth:** WFR provides faster energy dissipation than pure Wasserstein flows (see EDB in Appendix A.2), and may bypass the LSI constant limitation that plagues Langevin samplers in high dimensions, opening a new research direction for DRO.
>
> **Q2: Relationship to Xu et al. (2024) and Zhu & Xie (2024)**
>
>
> First, our main contribution is a principled PDE gradient flow framework. From our GF-DRO framework, deriving iterative optimization algorithms is just an exercise.
> *Xu et al. (FlowDRO)* learns a *deterministic* transport map via Neural ODEs with block-wise progressive training. That is NOT gradient flow. We differ fundamentally: (a) we *sample stochastically* from worst-case distributions via classical SDEs/PDEs — Xu et al. mention "sampling" but their mechanism is deterministic map-based, not stochastic sampling; (b) we handle entropy-regularized (Sinkhorn) DRO naturally and can easily recover WDRO, while FlowDRO only targets WDRO; we can also apply our framework to any DRO, with KL, MMD, as future works; (c) our convergence analysis uses classical PDE tools (LSI, displacement convexity), not neural network expressivity.
>
> *Zhu & Xie* analyze a modified JKO scheme where each inner iteration is a single proximal step. Our key conceptual contribution — the equivalence between the inner DRO problem and a sampling task (Proposition 3.1) — is absent from their work. This equivalence lets us leverage any gradient flow sampler (WGF, WFR, SVG). Moreover, their framework requires solving an infinite-dimensional optimization at each step, while ours reduces to practical particle simulation.
>
> We will move this comparison to the introduction.
>
> **Q3: Undefined notation**
>
> We will define all symbols at first use: $c$ (cost function) before Eq.(1); $V=-\ell$ before Eq.(2); $\tilde{V} _ {x,\tau}(y) \coloneqq V(y) + \frac{c(y,x)}{2\tau}$ before Example 1; $\pi_{Y|X}$ in Figure 1 caption. An expanded gradient flow primer will be added in the appendix for readers unfamiliar with gradient flows.
>
> **Q4: Technical issues**
>
> *Normalization in (29):* Will be added to match (7).
>
> *Proposition 4.2 vs. proof:* The reviewer is correct. The proof in (34) uses Kantorovich–Rubinstein duality, which requires the gradient oracle $g(y)=\nabla_\theta \ell(\theta,y)$ to be Lipschitz in $y$. The statement should read "$\nabla_\theta \ell(\theta, y)$ is $L$-Lipschitz in $y$." We will correct Proposition 4.2, Assumption 5.1(2), Theorem 5.2, and Lemma C.4 for consistency.
>
> *Randomness in (31):* Correct — $W_1(\hat{\rho}_Y, \pi_Y)$ is random. We fix this by taking expectations: $\mathbb{E}[W_1(\hat{\rho}_Y, \pi_Y)] \leq \mathbb{E}[W_1(\hat{\rho}_Y, \rho_t)] + W_1(\rho_t, \pi_Y)$, where the first term is the statistical error (vanishing as $N\to\infty$) and the second is the optimization error bounded by the gradient flow analysis. Eq.(31) and subsequent derivations will be updated.

---

> > ### Author Rebuttal · Reviewer_6af9 · 2026-04-04
> >
> > Thank you for the helpful clarification.

---

> > > ### Author Response · Authors · 2026-04-04
> > >
> > > Dear reviewer,
> > >
> > > Thank you for acknowledging the rebuttal.
> > >
> > > It seems that your concerns are fully addressed. In this case, could you please consider updating your score rating this reflect that? Is there any other clarification we can provide to improve the score?
> > >
> > > Many thanks,
> > > Authors

---

### Decision · Program_Chairs · 2026-04-30

**Decision:**

Accept (regular)

**Comment:**

This paper proposes a principled gradient-flow framework for DRO that recasts the inner worst-case distribution problem as sampling and unifies several existing methods under a common perspective. The main concerns were novelty relative to prior approaches, presentation,  and the strength of the empirical evidence. Overall, I find that the rebuttal addressed these points convincingly: the authors clarified the distinction from prior work, resolved or committed to fix the technical and notation issues, and strengthened the empirical case with additional experiments and a discussion of computational cost and scalability. Although some limitations remain, particularly regarding experimental scale, they do not outweigh the paper’s conceptual and technical contributions.